# REINFORCEMENT LEARNING OF DIVERSE SKILLS USING MIXTURE OF DEEP EXPERTS

## ABSTRACT

Agents that can acquire diverse skills to solve the same task have a benefit over other agents. Unexpected environmental changes for example may prohibit executing a learned behavior such that a complete retraining is necessary if the agent can not discard the invalid skill and rely on previously acquired, different ones. However, Reinforcement Learning (RL) policies mainly rely on Gaussian parameterization, preventing them from learning multi-modal, diverse skills. In this work, we propose a novel RL approach for training policies that exhibit diverse behavior. To this end, we propose a highly non-linear Mixture of Experts (MoE) as the policy representation, where each expert formalizes a skill as a contextual motion primitive. The context defines the task, which can be for instance the goal reaching position of the agent, or changing physical parameters like friction. Given a context, our trained policy first selects an expert out of the repertoire of skills and subsequently adapts the parameters of the contextual motion primitive. To incentivize our policy to learn diverse skills, we leverage a maximum entropy objective combined with a per-expert context distribution that we optimize alongside each expert. The per-expert context distribution allows each expert to focus on a context sub-space and boost learning speed. However, these distributions need to be able to represent multi-modality and hard discontinuities in the environment's context probability space. Moreover, the distributions should not rely on environmental pre-knowledge such as context boundaries, as they are usually not given. We solve these requirements by leveraging energy-based models to represent the per-expert context distributions and show how we can efficiently train them using the standard policy gradient objective. We show that our approach can learn precise and diverse skills of challenging robot simulation tasks.

## 1 INTRODUCTION

Recent advances in supervised policy learning have demonstrated the potential of training high-capacity policies capable of capturing multi-modal behaviors, as evidenced in recent studies (Shafiullah et al., 2022; Blessing et al., 2023; Chi et al., 2023). These policies have exhibited remarkably diverse skills and outperformed state-of-the-art methods. However, Reinforcement Learning (RL) policies usually rely on Gaussian parameterization that are able to discover only a single-mode solution to a task. While this limitation may suffice for tasks where environmental changes are not expected as for instance in production lines, achieving robustness in the face of dynamic environments, or learning adversarial strategies, such as playing table tennis against an opponent, requires agents to acquire diverse skills akin to human adaptability.

In this work, we propose a new approach for training policies that exhibit multi-modality within the behavioral space in the realm of RL. Our trained agents possess a diverse repertoire of skills from which they can select to tackle a specific task in different ways. We consider Contextual Reinforcement Learning in which a continuous-valued context defines the task (Kupcsik et al., 2013). A context can represent various scenarios, such as the location a robot needs to reach or varying physical parameters like friction or the desired position of an object or robot. Our method employs highly non-linear mixtures of expert policies to capture multi-modality within the action/behavior space of the agent. We also use automatic curriculum learning, enabling each expert to focus on a specific sub-region of the context space it favors. We introduce this curriculum shaping by optimizing for an additional per-expert context distribution that is used to sample contexts from the preferred regions

to train the corresponding expert. Automatic curriculum learning has proven to increase performance by improving the exploration of agents, particularly in sparse-rewarded environments (Klink et al., 2022). In the case of continuous context spaces, these distributions are often parameterized as Gaussian (Klink et al., 2020a; Celik et al., 2022). However, the agent is usually unaware of the context bounds, which makes additional techniques necessary to constrain the distribution updates to stay within the context region (Celik et al., 2022). Instead, we employ energy-based per-expert context distributions, which can be evaluated for any context and effectively represent multi-modality in the context space. Importantly, our model is trained solely using context samples from the environment that are inherently valid and within the defined bounds. This approach eliminates the need for additional regularization of the context distribution and does not require prior knowledge about the environment. Due to the overlapping probability distributions of different per-expert contexts, our resulting mixture policy offers diverse solutions for the same context. Recent research in RL has explored mixture of experts policies, but often these methods either train the mixture in unsupervised RL settings and then select the best-performing expert in the downstream task (Laskin et al., 2021; Eysenbach et al., 2019) or train linear experts, limiting their performance (Daniel et al., 2012; Celik et al., 2022). Our inspiration draws from recent advancements that have achieved diverse skill learning with a similar objective to ours. However, their approach involves linear expert models with Gaussian context distributions and requires prior knowledge of the environment to design a penalty term when the algorithm samples contexts outside of predefined bounds. These factors restrict the algorithm's performance and even its applicability if defining the context bounds need knowledge such as forward kinematics in robotics.

To summarize, in this paper, we introduce **Di-SkilL** – **Di**verse **Skil**l **L**earning, a novel RL method for learning a mixture of experts model. Our method is able to generalize to the continuous range of contexts defined by the environment's context distribution while learning multi-modal, and non-linear behaviors for solving a task defined by a specific context. Importantly, our approach operates without any assumption about the environment. To this end, we show how we can learn multi-modal context distributions by training an energy-based model solely on context samples obtained from the environment. Additionally, we demonstrate that we can learn high-performing and diverse behaviors on sophisticated simulated robotic tasks.

## 2 PRELIMINARIES AND RELATED WORK

**Contextual episode-based Policy Search (CEPS).** CEPS is a black-box approach to reinforcement learning (RL). In this framework, the search distribution is the agent's policy that is optimized for the mapping of contexts $\mathbf{c}$ to policy parameters, typically represented as motion primitives (Schaal, 2006; Paraschos et al., 2013; Li et al., 2023) parameterized by $\boldsymbol{\theta}$. The policy $\pi(\boldsymbol{\theta}|\mathbf{c})$ is optimized by

$$\max_{\pi(\boldsymbol{\theta}|\mathbf{c})} \mathbb{E}_{p(\mathbf{c})} \left[ \mathbb{E}_{\pi(\boldsymbol{\theta}|\mathbf{c})}[\mathrm{R}(\mathbf{c}, \boldsymbol{\theta})] \right], \tag{1}$$

where $\mathrm{R}(\mathbf{c}, \boldsymbol{\theta})$ is the return. Given context samples from the environment's context distribution $p(\mathbf{c})$, the policy $\pi(\boldsymbol{\theta}|\mathbf{c})$ chooses the controller's parameters $\boldsymbol{\theta}$ once in the beginning of the episode. One of the noteworthy advantages of contextual episode-based RL lies in the independence of assumptions such as the Markovian property in common MDPs. This characteristic renders it a versatile methodology, particularly well-suited for addressing a diverse array of intricate tasks where the formulation of a Markovian reward function proves elusive. For instance, it demonstrates particular efficacy in scenarios demanding the retrospective evaluation of an agent's performance, such as in tasks involving the rewarding of an agent based on its maximum achieved height, as encountered in jumping tasks (Otto et al., 2023). CEPS has been explored by researchers who have applied various optimization techniques, including Policy Gradients (Sehnke et al., 2010), Natural Gradients (Wierstra et al., 2014), stochastic search strategies (Hansen & Ostermeier, 2001; Mannor et al., 2003; Abdolmaleki et al., 2019), and trust-region optimization techniques (Abdolmaleki et al., 2015; Daniel et al., 2012; Tangkaratt et al., 2017), particularly in the non-contextual setting. Researchers have expanded the scope of these settings by incorporating linear contextual adaptation (Tangkaratt et al., 2017; Abdolmaleki et al., 2019) as well as non-linear adaptation (Otto et al., 2023), leveraging the recently introduced trust-region layers for neural networks (Otto et al., 2021). All of the previously mentioned methods focus on learning single-mode policies and do not address acquiring diverse skills leveraging automatic curriculum learning, which are key aspects that distinguish our research.

**Curriculum Reinforcement Learning.** CRL has the potential to increase the performance of RL agents, especially in sparse-rewarded environments in which exploration is fundamentally difficult. Adapting the environment based on the agent's learning process has been proposed by several works already, e.g. automatically generating sets of tasks or goals to increase the learning speed of the agent (Florensa et al., 2017; Sukhbaatar et al., 2018; Zhang et al., 2020; Florensa et al., 2018), or generating a curriculum by interpolating an auxiliary and a known distribution of target tasks (Klink et al., 2022; 2020a;b). Other works propose sampling a training level from a prespecified set of environments (Jiang et al., 2021b), or design an environment in an unsupervised manner (Jiang et al., 2021a; Dennis et al., 2020) based on the agent's learning process. None of the aforementioned methods apply automatic curriculum learning on a RL problem with an MoE policy, except for the work in (Celik et al., 2022). They, however, parameterize the curriculum distribution as a Gaussian where we consider an energy-based model which has many benefits as we show in Section 3.

**Mixture of Experts (MoE) Policy for Curriculum Learning.** The MoE policy is formalized as

$$\pi(\boldsymbol{\theta}|\mathbf{c}) = \sum_o \pi(o|\mathbf{c})\pi(\boldsymbol{\theta}|\mathbf{c}, o), \tag{2}$$

where the gating distribution $\pi(o|\mathbf{c})$ assigns an expert $o$ to the given context $\mathbf{c}$. The expert $\pi(\boldsymbol{\theta}|\mathbf{c}, o)$ adapts the parameters $\boldsymbol{\theta}$ of the motion primitive for $\mathbf{c}$. The corresponding motion primitive is then executed in the environment. While this form of the MoE is suitable in inference time where the context is assigned by the environment and the agent needs to propose a skill, it does not allow to automatically learn a curriculum during training. This drawback is caused by the lack of a parameterized distribution $\pi(\mathbf{c})$ that is part of the MoE and allows to explicitly choose and set context samples for the model itself such that each expert can decide on which contexts it favors training. Introducing a generative model in the context space is a small, but necessary distinction to enable automatic curriculum learning for each single expert $o$. We can easily reparameterize the MoE without any assumption by using Bayes' rule as Celik et al. (2022)

$$\pi(\boldsymbol{\theta}|\mathbf{c}) = \sum_o \frac{\pi(\mathbf{c}|o)\pi(o)}{\pi(\mathbf{c})}\pi(\boldsymbol{\theta}|\mathbf{c}, o). \tag{3}$$

The per-expert context distribution $\pi(\mathbf{c}|o)$ can now be optimized and allows the expert $o$ to choose contexts $\mathbf{c}$ it favors. Note that $\pi(\mathbf{c}) = \sum_o \pi(\mathbf{c}|o)\pi(o)$. We model each $\pi(\mathbf{c}|o)$ as an energy-based model and each $\pi(\boldsymbol{\theta}|\mathbf{c}, o)$ as a Gaussian parameterized as a neural network (see also Fig. 1). The prior $\pi(o)$ is set to be a uniform distribution throughout this work.

**Self-Paced Diverse Skill Learning with Mixture of Experts (MoE).** Discovering different skills in the same context-defined task is called learning diverse skills. MoE models (see Eq. 3) are specifically suitable for skill discovery due to their ability to represent multi-modality and the per-expert context distribution $\pi(\mathbf{c}|o)$ for automatic curriculum learning which allows the experts to specialize in a sub-set of the context space. For explicit optimization of the aforementioned properties, the KL-regularized Maximum Entropy Reinforcement Learning objective (Celik et al., 2022)

$$\max_{\pi(\boldsymbol{\theta}|\mathbf{c}),\pi(\mathbf{c})} \mathbb{E}_{\pi(\mathbf{c})}\left[\mathbb{E}_{\pi(\boldsymbol{\theta}|\mathbf{c})}[\mathrm{R}(\mathbf{c},\boldsymbol{\theta})] + \alpha\mathrm{H}[\pi(\boldsymbol{\theta}|\mathbf{c})]\right] - \beta\mathrm{KL}(\pi(\mathbf{c}) \| p(\mathbf{c})) \tag{4}$$

is a natural choice. The KL-term in the objective allows for curriculum learning in which the context distribution $\pi(\mathbf{c})$ is optimized to match the environment's distribution $p(\mathbf{c})$. This part of the objective can be prioritized during optimization by choosing the scaling parameter $\beta$ appropriately. The entropy of the mixture model incentivizes learning versatile solutions (Celik et al., 2022) and can be prioritized with a high scaling parameter $\alpha$. Inserting $\pi(\boldsymbol{\theta}|\mathbf{c})$, $\pi(\mathbf{c})$ from Eq. (3) into Eq. (5) and applying Bayes theorem leads to

$$\max_{\pi(\mathbf{c},\boldsymbol{\theta})} \mathbb{E}_{\pi(o),\pi(\mathbf{c}|o)}\left[\mathbb{E}_{\pi(\boldsymbol{\theta}|\mathbf{c},o)}[\mathrm{R}(\mathbf{c},\boldsymbol{\theta}) + \alpha\log\pi(o|\mathbf{c},\boldsymbol{\theta})] + \beta\log p(\mathbf{c}) + (\beta-\alpha)\log\pi(o|\mathbf{c})\right]$$
$$+ \alpha\mathbb{E}_{\pi(o),\pi(\mathbf{c}|o)}[\mathrm{H}[\pi(\boldsymbol{\theta}|\mathbf{c},o)]] + \beta\mathbb{E}_{\pi(o)}[\mathrm{H}[\pi(\mathbf{c}|o)]] + \beta\mathrm{H}[\pi(o)] \tag{5}$$

It is well-known that this objective is difficult to optimize for MoE policies and requires further steps to obtain a per-component lower-bound Celik et al. (2022)

$$\max_{\pi(\boldsymbol{\theta}|\mathbf{c},o)} \mathbb{E}_{\pi(\mathbf{c}|o),\pi(\boldsymbol{\theta}|\mathbf{c},o)}[\mathrm{R}(\mathbf{c},\boldsymbol{\theta}) + \alpha\log\tilde{\pi}(o|\mathbf{c},\boldsymbol{\theta})] + \alpha\mathbb{E}_{\pi(\mathbf{c}|o)}[\mathrm{H}[\pi(\boldsymbol{\theta}|\mathbf{c},o)]] \tag{6}$$

for the expert updates and a per-component lower-bound for the per-expert context updates

$$\max_{\pi(\mathbf{c}|o)} \mathbb{E}_{\pi(\mathbf{c}|o)} \left[ L_c(o, \mathbf{c}) + (\beta - \alpha) \log \tilde{\pi}(o|\mathbf{c}) \right] + \beta \mathrm{H}\left( \pi(\mathbf{c}|o) \right), \tag{7}$$

where $L_c(o, \mathbf{c}) = \mathbb{E}_{\pi(\boldsymbol{\theta}|\mathbf{c}, o)} \left[ \mathrm{R}(\mathbf{c}, \boldsymbol{\theta}) + \alpha \log \tilde{\pi}(o|\mathbf{c}) \right] + \alpha \mathrm{H} \left[ \pi(\boldsymbol{\theta}|\mathbf{c}, o) \right]$. The variational distributions $\tilde{\pi}(o|\mathbf{c}, \boldsymbol{\theta}) = \pi_{old}(o|\mathbf{c}, \boldsymbol{\theta})$ and $\tilde{\pi}(o|\mathbf{c}) = \pi_{old}(o|\mathbf{c})$ arise through the decomposition and are responsible for learning diverse solutions and concentrating on context regions with small, or no support by $\pi(\mathbf{c})$. Every iteration, the variational distributions are updated to tighten the bounds. The exact derivations can be found in (Celik et al., 2022).

**Diverse Skill Learning.** Ren et al. (2021) proposes using MoE policy representation and presents a novel gradient estimator to calculate the gradients w.r.t. the MoE parameters. (Huang et al., 2023) presents a model-based RL approach to train latent variable models. The work presents a novel lower bound for training the multi-modal policy parameterization. These methods differ to our work in that they are not categorized in the CEPS framework and do not use automatic curriculum learning techniques. In the CEPS framework, Diverse Skill Learning with MoE models has also been explored in the works by Daniel et al. (2012); End et al. (2017). They, however, consider learning an MoE model with linear experts without automatic curriculum learning and need to add additional constraints to enforce diversity in the experts. The work by Celik et al. (2022) also relies on the maximum entropy objective as we do, however, their method only considers linear experts with Gaussian per-expert distributions which limits the performance and consequently requires many experts to solve a task. Moreover, it requires environment knowledge to hand-tune a punishment term to keep the optimization of the per-expert context distributions within the context bounds.

**Unsupervised Reinforcement Learning.** Another field of research that considers learning diverse policies is unsupervised reinforcement learning (URL). In URL the agent is first trained solely with an intrinsic reward to acquire a diverse set of skills from which the most appropriate is picked to solve a downstream task. More related to our work is a group of algorithms that obtain their intrinsic reward based on information-theoretic formulations (Laskin et al., 2021; Eysenbach et al., 2019; Campos et al., 2020; Lee et al., 2019; Liu & Abbeel, 2021). However, their resulting objective is based on the mutual-information and differs from the objective we maximize. The learned skills in the pre-training aim to cover distinct parts of the state-space during pre-training in the absence of an extrinsic task reward which implies that skills are not explicitly trained to solve the same task in different ways. Those methods operate within the step-based RL setting which differs from CEPS.

## 3 DIVERSE SKILL LEARNING

In this section, we present Di-SkilL. We provide a high-level overview of our method, discuss emerging issues and show how we address them. In particular, we show how we can use energy-based models for automatic curriculum learning.

### 3.1 HIGH-LEVEL OVERVIEW OF DI-SKILL

The common Contextual Episodic Policy Search (CEPS) (Kupcsik et al., 2013) with a Mixture of Experts (MoE) policy representation learning loop observes a context $\mathbf{c}$, selects an expert $o$ that subsequently adjusts the controller parameters $\boldsymbol{\theta}$ given $(\mathbf{c}, o)$. We consider the same process during testing time as shown in blue color in Fig. 1 and in the corresponding graphical model in Fig. 2a. However, the procedure changes during training for Di-SkilL as automatic curriculum learning requires that the agent can determine which context regions it prefers to focus on. In this case, we observe a batch of context samples from the environment's context distribution $p(\mathbf{c})$. For each of these samples, every per-expert context distribution $\pi(\mathbf{c}|o)$ calculates a probability, which results in a categorical distribution. We use these probabilities to sample contexts for each expert $\pi(\boldsymbol{\theta}|\mathbf{c}, o)$ resulting in $(\mathbf{c}, o)$ samples since this sampling is repeated for each expert $o$. The training is illustrated in orange color in Fig. 1 and shown in the graphical model in Fig. 2b. Each chosen expert $o$ provides a Gaussian distribution over the motion primitive parameters $\boldsymbol{\theta}$ by mapping the context $\mathbf{c}$ to a mean vector $\boldsymbol{\mu_\theta}$ and a covariance matrix $\boldsymbol{\Sigma_\theta}$. A trajectory $\tau$ is generated and subsequently executed by a trajectory following controller on the environment by providing the motion primitive generator a sampled parameter $\boldsymbol{\theta}$. The trajectory generation and execution process is visualized in green color in Fig. 1. For each $(\mathbf{c}, o)$ sample, the agent observes an episode return $\mathrm{R}(\mathbf{c}, \boldsymbol{\theta})$ which is used for

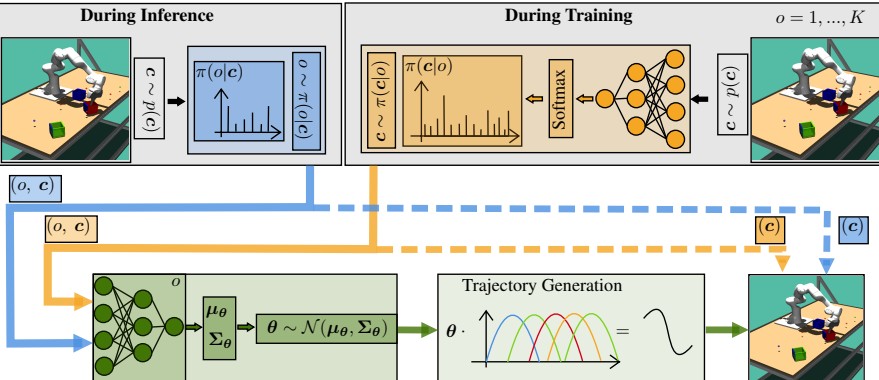

Figure 1: **The Sampling procedure for Di-SkilL.** During **Inference** the agent observes contexts **c** sampled from the environment's unknown context distribution $p(\mathbf{c})$. The agent calculates the gating probabilities $\pi(o|\mathbf{c})$ for each context and samples an expert $o$ resulting in $(o, \mathbf{c})$ samples marked in blue. During **Training** we first sample a batch of contexts **c** from $p(\mathbf{c})$. We use this batch to calculate the distribution $\pi(\mathbf{c}|o)$ for each individual expert $o = 1, ..., K$. The per-expert context distribution $\pi(\mathbf{c}|o)$ provides higher probability for contexts that are preferred by the expert $\pi(\boldsymbol{\theta}|\mathbf{c}, o)$. To enable curriculum learning, we provide each expert the contexts sampled from its corresponding $\pi(\mathbf{c}|o)$, resulting in the samples $(o, \mathbf{c})$ marked in orange. For both procedures, the chosen $\pi(\boldsymbol{\theta}|\mathbf{c}, o)$ samples motion primitive parameters $\boldsymbol{\theta}$ for each **c** resulting in a trajectory $\tau$ that is subsequently executed on the environment. Before execution, the corresponding context **c**, e.g., goal position of a box needs to be set in the environment. This is illustrated by the dashed arrows with the corresponding context in blue for inference and orange for training.

updating the MoE as we show in Section 3.3. Yet, there exist several issues for a stable overall training of the MoE model, which requires special treatment for each $\pi(\mathbf{c}|o)$ and $\pi(\boldsymbol{\theta}|\mathbf{c}, o)$. We showcase and address them in the following sections.

## 3.2 ENERGY-BASED MODEL FOR AUTOMATIC CURRICULUM LEARNING

Fig. 2d illustrates a two-dimensional environment's context distribution $p(\mathbf{c})$. Even though only a uniform distribution, it is challenging for the Reinforcement Learning (RL) agent to automatically learn its curriculum $\pi(\mathbf{c}|o)$ within the valid context space due to the following reasons. Hard discontinuities such as steps often naturally arise in $p(\mathbf{c})$ due to the environment's finite support. For instance, in an environment where the agent's task is to place an object to specific positions on a table, the probability of observing a goal position outside the table's surface is zero. This implies that a large subset of the context space has no probability mass. Therefore, exploration in these regions might be difficult if there is no guidance encoded in the reward. Even if it is guaranteed that $\pi(\mathbf{c}|o)$ only samples valid contexts, it still needs to be able to represent multi-modal distributions, such as illustrated in Fig. 2d. Multi-modality can easily occur if experts $\pi(\boldsymbol{\theta}|\mathbf{c}, o)$ prefer contexts in spatially apart regions. Because of these reasons we require $\pi(\mathbf{c}|o)$ to be able to represent **i)** complex distributions, **ii)** multi-modality and **iii)** only explore within the valid context bounds of $p(\mathbf{c})$. We propose parameterizing $\pi(\mathbf{c}|o)$ as an energy-based model (EBM)

$$\pi(\mathbf{c}|o) = \frac{\exp(\boldsymbol{\phi}_o(\mathbf{c}))}{Z} \tag{8}$$

to address the issues i) and ii). EBMs have shown to be capable of representing sharp discontinued functions and multi-modal distributions (Florence et al., 2022). Yet, they are hard to train and sample from due to the intractable normalizing constant $Z = \int_{\mathbf{c}} \exp(\boldsymbol{\phi}_o(\mathbf{c}))d\mathbf{c}$. We can easily circumvent these issues and additionally address issue iii) by approximating the normalizing constant with contexts $\mathbf{c} \sim p(\mathbf{c})$ as $Z \approx \frac{1}{N} \sum_{i=1}^{N} \exp(\phi_o(\mathbf{c}_i))$. This approximation is justified as we can easily sample from $p(\mathbf{c})$ by simply resetting the environment without execution. Additionally, by resampling a large enough batch of contexts $\mathbf{c} \sim p(\mathbf{c})$ in each iteration, the EBM will encounter important parts of the context space during the training. Each expert can therefore sample preferred contexts from the current batch of valid contexts by simply calculating the probability for each of the contexts using $\pi(\mathbf{c}|o)$ as parameterized in Eq. 8. Note that this sampling procedure is not straightforwardly

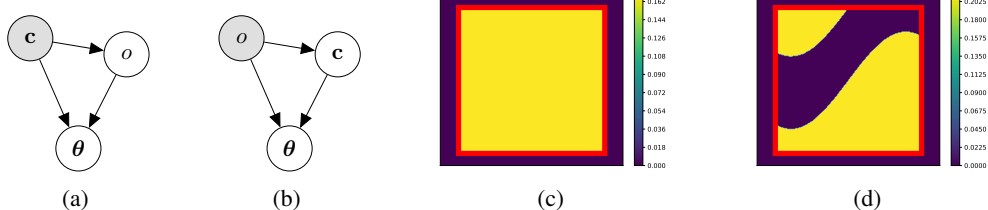

(a)        (b)        (c)        (d)

Figure 2: Probabilistic Graphical Models (PGMs) during **inference a)** and **training b)**. During **a))** the model observes the contexts **c** from the environment. An expert $o$ is sampled from $\pi(o|\mathbf{c})$, which subsequently leads to an adjustment of the motion primitive parameters by $\pi(\boldsymbol{\theta}|\mathbf{c}, o)$. We iterate over each expert during **(b)**, sample the contexts **c** and the motion primitive parameter $\boldsymbol{\theta}$ from the per-expert distribution $\pi(\mathbf{c}|o)$ and $\pi(\boldsymbol{\theta}|\mathbf{c}, o)$ respectively. Sampling from $\pi(\mathbf{c}|o)$ allows shaping the expert's curriculum. **c)** illustrates the environment's context distribution $p(\mathbf{c})$ **(c))** and a possibly optimal $\pi(\mathbf{c}|o)$ **(d))** in two-dim. space. Yellow areas indicate high and purple zero probability. The illustrations show that optimizing $\pi(\mathbf{c}|o)$ requires dealing with i) step-like non-linearities, ii) multi-modality, iii) bounded within the red rectangle support of $p(\mathbf{c})$, complicating exploration.

applicable to explicit models such as Gaussians, or Normalizing Flows (Papamakarios et al., 2021). Those methods would need additional techniques like importance sampling that might destabilize learning if not carefully calibrated by enforcing overlapping support regions of the sampling and the actual distribution. In our case, updating the parameters of the EBM can easily be addressed by the standard RL objective for diverse skill learning, as we show in the next sections.

### 3.3    UPDATING THE MIXTURE OF EXPERTS MODEL

We update each expert $\pi(\boldsymbol{\theta}|\mathbf{c}, o)$ and its corresponding per-expert context distribution $\pi(\mathbf{c}|o)$ by maximizing the objectives in Eq. 6 and in Eq. 7 respectively. These decomposed objectives allow to independently update both distributions and to retain the properties of diverse skill learning from the objective in Eq. 5. However, updating the distributions is not straightforward due to the bi-level optimization that leads to a dependency of both terms. This is in particular problematic for the expert $\pi(\boldsymbol{\theta}|\mathbf{c}, o)$ as the sampled contexts **c** can drastically change from one iteration to another if $\pi(\mathbf{c}|o)$ changes too aggressively. The same applies for updating $\pi(\mathbf{c}|o)$ as calculating the objective requires calculating an integral over $\boldsymbol{\theta}$ under the expectation of $\pi(\boldsymbol{\theta}|\mathbf{c}, o)$. For a stable update for both distributions, we employ trust-region updates to restrict the change of both distributions from an iteration to another. Trust-Region updates have shown to considerably improve the learning progress recently (Otto et al., 2021; Schulman et al., 2015; 2017).

**Expert Update.**    We parameterize each expert $\pi(\boldsymbol{\theta}|\mathbf{c}, o)$ with a single neural network and update them by a trust-region constrained optimization

$$\max_{\pi(\boldsymbol{\theta}|\mathbf{c},o)} \mathbb{E}_{\pi(\mathbf{c}|o),\pi(\boldsymbol{\theta}|\mathbf{c},o)}\left[\mathrm{R}(\mathbf{c},\boldsymbol{\theta}) + \alpha \log \tilde{\pi}(o|\mathbf{c},\boldsymbol{\theta})\right] + \alpha \mathbb{E}_{\pi(\mathbf{c}|o)}\left[\mathrm{H}\left[\pi(\boldsymbol{\theta}|\mathbf{c},o)\right]\right]$$

$$\text{s.t.} \qquad \mathrm{KL}\left(\pi(\boldsymbol{\theta}|\mathbf{c},o) \parallel \pi_{\text{old}}(\boldsymbol{\theta}|\mathbf{c},o)\right) \leq \epsilon \quad \forall \mathbf{c} \in \mathcal{C}, \tag{9}$$

where the KL-bound ensures that the expert $\pi(\boldsymbol{\theta}|\mathbf{c}, o)$ does not differ too much from the expert $\pi_{\text{old}}(\boldsymbol{\theta}|\mathbf{c}, o)$ from the iteration before. We efficiently update the experts using trust region layers Otto et al. (2021; 2023). The entropy bonus incentivizes $\pi(\boldsymbol{\theta}|\mathbf{c}, o)$ to fully cover the parameter space, while avoiding $(\boldsymbol{\theta}, \mathbf{c})$ regions that are covered by other experts $o$. The latter is guaranteed by $\tilde{\pi}(o|\mathbf{c}, \boldsymbol{\theta})$ which rewards $(\boldsymbol{\theta}, \mathbf{c})$ regions that can be assigned to expert $o$ with high probability.

**Per-Expert Context Distribution Objective.**    We consider the objective with the augmented rewards as shown in Eq. 7 for updating each $\pi(\mathbf{c}|o)$ distribution. We can not apply the trust region layers (Otto et al., 2021) in this case, as $\pi(\mathbf{c}|o)$ is a discrete distribution parameterized by the EBM. Yet, we can still use PPO (Schulman et al., 2017) for updating $\pi(\mathbf{c}|o)$ and simplify our objective, as we can now calculate many terms in closed form. For this, we rewrite the objective as

$$\max_{\pi(\mathbf{c}|o)} \sum \pi(\mathbf{c}|o)L_c(o,\mathbf{c}) + \sum_i \pi(\mathbf{c}|o)\left((\beta - \alpha)\left(\log \tilde{\pi}(o|\mathbf{c}) - \log \sum_o \tilde{\pi}(o|\mathbf{c})\right) - \beta \log \pi(\mathbf{c}|o)\right)$$

$$\tag{10}$$

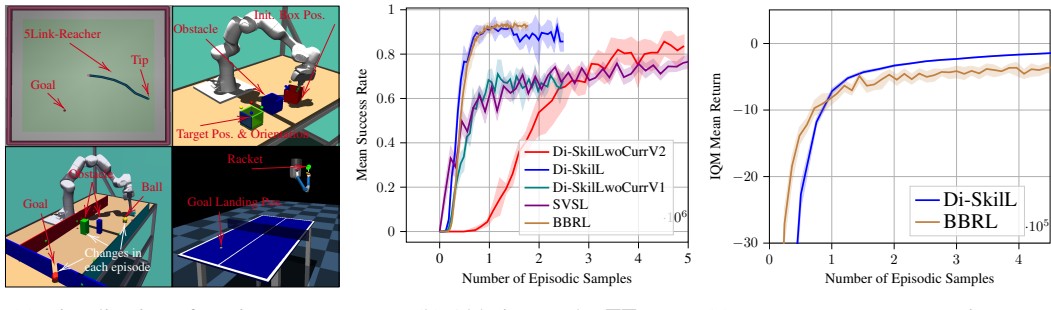

(a) Visualization of Environments    (b) Ablation on the-**TT**    (c) Mean Return on RT Environment

Figure 3: **a)** (left top) Reacher Task (**RT**). In RT, a 5-Link Reacher has to reach a goal with its tip. The context space is the 2-dim. position of the goal in the XY-surface. (Top right) Box Pushing (**BP**). In (**BP**) a 7DoF robot has to push the red box to the target position (green) while avoiding the obstacle (blue), where the blue sides of the boxes need to align. (Bottom Left) mini golf (**MG**). In (**MG**) a 7DoF robot has to hit the ball such that it passes through the tight goal while avoiding the obstacles. The context space is the 2-dim. position of the obstacle, the X-positions of the ball and the goal (total 4dim.). (Bottom right) table tennis (**TT**). In the (**TT**) environment a 7DoF robot has to return a ball to a desired ball landing position. The context consists of the 2-dim. ball serving position and the 2-dim. desired goal position. In the more complex version we increase the context dim. to five by including varying initial ball velocities. **b)** Ablation studies, showcasing the need of automatic curriculum learning for Di-SkilL. BBRL and Di-SkilL can solve the four-dim. TT, where its variants without curriculum learning (Di-SkilLwoCurV1, Di-SkilLwoCurrV2) struggle to achieve a good performance. SVSL needs more samples to achieve around 80% success rate, suffering under the linear experts. **c) Performance of Di-SkilL and BBRL on RT.**

and observe that all terms in the second sum can be calculated in closed-form. Note that the first term is approximated by samples from $\pi(\mathbf{c}|o)$ since it requires calculating the integral over $\boldsymbol{\theta}$ under the expectation of $\pi(\boldsymbol{\theta}|\mathbf{c}, o)$ because of $L_c(o, \mathbf{c}) = \mathbb{E}_{\pi(\boldsymbol{\theta}|\mathbf{c},o)}\big[\mathrm{R}(\mathbf{c}, \boldsymbol{\theta}) + \alpha \log \tilde{\pi}(o|\mathbf{c})\big] + \alpha\mathrm{H}\left[\pi(\boldsymbol{\theta}|\mathbf{c}, o)\right]$. The entropy bonus in Eq. (10) incentivizes to cover the context space, while focusing on context regions that are not, or only partly coverd by other options. The latter is guaranteed by $\tilde{\pi}(o|\mathbf{c})$ which assigns a high probability if expert $o$ can be assigned to the context $\mathbf{c}$.

## 4 EXPERIMENTS

In our empirical evaluations, we compare our method against the baselines BBRL (Otto et al., 2023) and SVSL (Celik et al., 2022). Both methods are suitable baselines as they are state of the art algorithms in the field of Contextual Episode-Based Policy Search (CEPS). BBRL is able to learn highly non-linear policies leveraging trust region updates. SVSL learns linear Mixture of Experts (MoE) models and is able to capture multi-modality in the behavior space. We aim to clarify how important the automatic curriculum learning for Di-SkilL is and whether Di-SkilL is able to learn **high-performing** and **diverse** skills. We consider challenging robotic environments with continuous context and parameter spaces. The considered environments either have a non-markovian, i.e. requires retrospect data for calculation, or temporally sparse reward functions which additionally increases the learning complexity. Note that we use ProDMPs (Li et al., 2023) to generate trajectories throughout our environments. Experimental details can be found in the Appendix C.

### 4.1 DO WE NEED AUTOMATIC CURRICULUM LEARNING?

An important feature of Di-SkilL is that each expert is able to shape its own curriculum by explicitly sampling from preferred context regions and gradually increasing the covered context space with increasing performance. We show the importance of this feature by disabling the automatic curriculum learning, by setting $\log \tilde{\pi}(o|\mathbf{c}) = 0$ in Eq. 10 and setting the entropy scaling parameter $\beta = 2000$ to a very high value such that $\pi(\mathbf{c}|o)$ is uniformly distributed over the context space. Setting $\log \tilde{\pi}(o|\mathbf{c}) = 0$ eliminates the intrinsic motivation of each $\pi(\mathbf{c}|o)$ to focus on sub-regions in the context space that are not, or only partially, covered by any other per-expert distribution. We evaluate two variants of Di-SkilL. The variational distribution is set to zero and beta is to $\beta = 2000$ in both

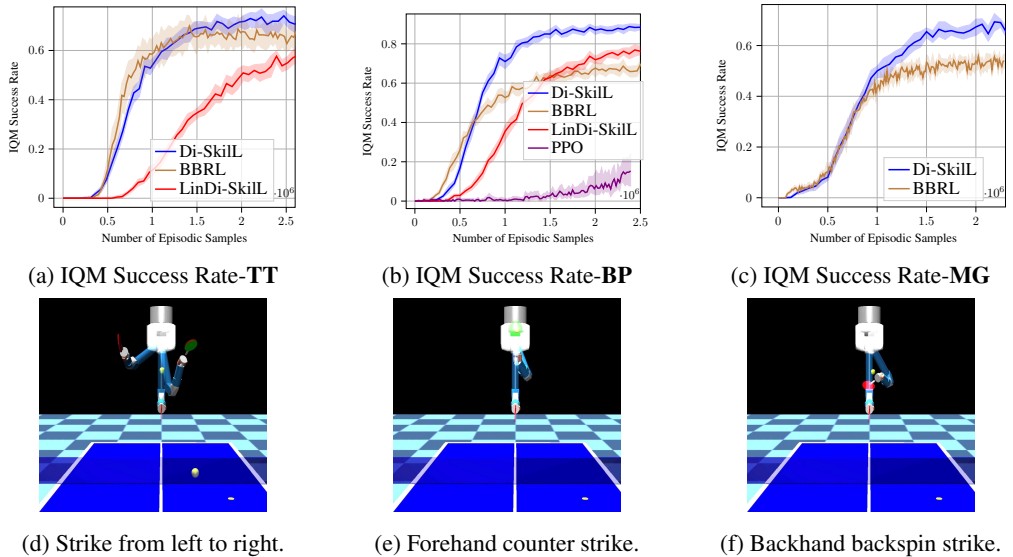

(a) IQM Success Rate-**TT**     (b) IQM Success Rate-**BP**     (c) IQM Success Rate-**MG**

(d) Strike from left to right.     (e) Forehand counter strike.     (f) Backhand backspin strike.

Figure 4: **Performance on the extended a) TT, b) BP and c) MG tasks. a)** While BBRL converges faster, Di-SkilL achieves a higher success rate eventually. **b)** The multi-modality introduced by the obstacle in the box pushing task leads to around 65% for BBRL and around 85% succes rate for Di-SkilL. Di-SkilL is able to represent multi-modality in the context **c** and parameter $\theta$ space. **c)** Di-SkilL achieves around 20% higher success rate on the MG task.

variants. For Di-SkilLwoCurV1, we provide the same number of 50 context-parameter samples per expert as in Di-SkilL, whereas Di-SkilLwoCurV2 receives 260 samples per expert in each iteration. All variants of Di-SkilL consist of five experts. Note that $\beta = 0.5$ for Di-SkilL as it showcases our method with all its inherent features. We run all methods on the table tennis environment, in which a 7DoF robot has to learn fast and precise motions to smash the ball on the desired position on the opponent's side (see Fig. 3a and Appendix C) (Otto et al., 2023). A strike is considered as successful if the distance of the ball's landing position and the goal is smaller than 0.2m. The table tennis environment requires good exploratory behavior and has a non-markovian reward structure, which makes state of the art step-based approaches infeasible to learn useful skills (Otto et al., 2023). Fig. 3b shows the mean success rates and the 95% confidence interval for each method on at least four seeds. BBRL and Di-SkilL achieve a very high success rate. However, we can clearly see that Di-SkilLwoCurV1 converges to a much smaller success rate and Di-SkilLwoCurV2 needs much more samples to reach the level of Di-SkilL. Interestingly, SVSL also shows worse performance, even though the model has 20 experts. The results show that automatic curriculum learning is a necessary feature for Di-SkilL to solve the task and that linear experts are not capable of achieving a satisfying performance. SVSL requires desing a punishment function to guide the context samples in the valid context region, which makes its application difficult, especially if the context influences the objects' physics. We therefore propose comparing to BBRL and LinDi-SkilL instead of SVSL. LinDi-SkilL benefits from Di-SkilL's energy-based $\pi(\mathbf{c}|o)$ and hence does not require additional treatment, but has linear expert parameterizations as SVSL.

### 4.2 ANALYZING THE PERFORMANCE AND DIVERSITY OF SKILLS

We consider more complex variants of the 5-Link Reacher, table tennis and the box pushing environment introduced by (Otto et al., 2023). Additionally, we benchmark on the robot minigolf environment (see Fig. 3a and Appendix C for details). We report the performances of Di-SkilL, Lin-DiSkill and BBRL, and analyze the learned diverse solutions of Di-SkilL. We have conducted 24 seeds for each environment and algorithm and report the *interquartile mean* (IQM) with a 95% stratified bootstrap confidence interval as suggested by Agarwal et al. (2021).

**5-Link Reacher Environment (5LRE).** Initially in (Otto et al., 2023) only the first and second quadrant were considered as goal reaching position to avoid multi-modal solutions. We consider all quadrants and compare to BBRL in Fig. 3c. Di-SkilL converges a bit slower than BBRL, but eventually achieves a higher return.

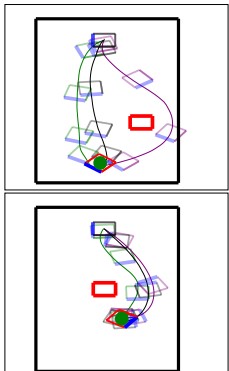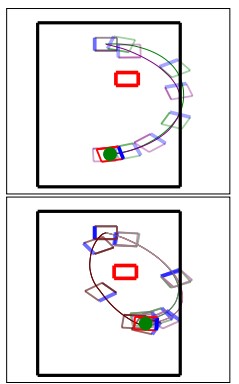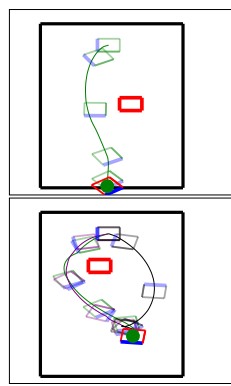

Figure 5: **Di-SkilL's Diverse Skills for the BP Task.** The figures visualize diverse solutions to the same contexts **c** on a table (black rectangle). The red, thick rectangle represents the obstacle. The 7DoF robot is tasked to push the box (shown in different colors for each solution found) to the goal box position (red rectangle with a green dot) and align the blue edges to match the orientation. We visualized successful box trajectories for each sampled skill. The diversity learned in the parameter space results in different box trajectories ranging in the position and the orientations.

**Table Tennis Environment (TTE).** We extend the TTE by varying the initial ball velocity in the agent's direction during the serve. This additionally increases the learning complexity, as the agent now needs to reason about the physical effects of changed velocity ranges. The performance can be seen in Fig. 4a. Di-SkilL achieves similar performance as BBRL, but eventually surpasses BBRL's success rate slightly. Both methods show less success rate than for the easier variant (Section 4.1). Yet, Di-SkilL is able to learn diverse skills to the same or similar contexts, as visualized in Fig. 4.

**Box Pushing Environment (BPE).** The 7DoF Robot has to push a box to a target position and rotation on a table while avoiding an obstacle in an increased context region. A box push is considered successful if the distance of the box' and the target box position is smaller than 5cm and the z-axis orientation error is smaller than 0.5rad. Fig. 4b shows the success rate of BBRL and Di-SkilL. Di-SkilL outperforms BBRL with a success rate of around 70% to 50%. The obstacle introduces multi-modality in the behavior space which can not be captured by a single-mode policy, explaining BBRL's low success rate. Fig. 5 shows different box trajectories learned by Di-SkilL.

**Robot Minigolf Environment (MGE).** In the MGE the robot has to hit the ball in a wall-surrounded environment with two obstacles, such that it passes through the tight goal by at least 0.75m. Note that one of the obstacles' position is resampled in each episode. The MGE is a challenging environment as the agent has to infer the ball's trajectory and possible collisions with the obstacles and the walls while precisly hitting the ball through the tight whole. Fig. 4c shows the performances of Di-SkilL and BBRL. Di-SkilL achieves a success rate of around 70% while BBRL converges to around 50%, showing that Di-SkilL is able to learn precise solutions and overcome possible multi-modalities.

## 5    CONCLUSION

We proposed a novel method for learning diverse skills using contextual Mixture of Deep Experts. Each expert automatically learns its curriculum by optimizing for a per-expert context distribution $\pi(\mathbf{c}|o)$. We have demonstrated major challenges which arise through enabling automatic curriculum learning (ACR) and proposed parameterizing $\pi(\mathbf{c}|o)$ as energy-based models (EBMs) to address these challenges. Additionally, we provided a methodology to efficiently optimize these EBMs. We also proposed using trust-region updates for the deep experts for stabilizing our bi-level optimization problem. In an ablation we have shown that ACR is necessary for efficient and performant learning. Moreover, on sophisticated robot simulation environments, we have shown that our method outperforms the baselines and learns diverse skills. Currently, the major drawback of our approach is that it is not able to replan, causing failures in the tasks if the robot even has small collisions with objects. We intend to address this issue in future research. Additionally, techniques such as intra-option learning might reduce the sample-complexity.

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

# A   The Parameterization of the Mixture of Experts (MoE) Model

In the following we provide details on the parameterization of the MoE model.

**Parametrization of the expert** $\pi(\boldsymbol{\theta}|\mathbf{c}, o)$. We parameterize each expert $\pi(\boldsymbol{\theta}|\mathbf{c}, o)$ as a Gaussian policy $\mathcal{N}(\boldsymbol{\mu}_{\boldsymbol{\gamma}}(\mathbf{c}), \boldsymbol{\Sigma}_{\boldsymbol{\gamma}}(\mathbf{c})$, where the mean $\boldsymbol{\mu}_{\boldsymbol{\gamma}}(\mathbf{c})$ and the covariance $\boldsymbol{\Sigma}_{\boldsymbol{\gamma}}(\mathbf{c})$ are functions of the context $\mathbf{c}$ and parameterized by a neural network with parameters $\boldsymbol{\gamma}$. Although the covariance $\boldsymbol{\Sigma}_{\boldsymbol{\gamma}}(\mathbf{c})$ is formalized as a function of the context $\mathbf{c}$, we have not observed any advantages in doing so. In our experiments we therefore parameterize the covariance as a lower-triangular matrix $\mathbf{L}$ and form the covariance matrix $\boldsymbol{\Sigma} = \mathbf{L}\mathbf{L}^T$.

**Parameterization of the per-expert context distribution** $\pi(\mathbf{c}|o)$. The reader is referred to Section 3 for details on the parameterization of $\pi(\mathbf{c}|o)$

**Parameterization of the prior** $\pi(o)$. We fix the prior $\pi(o)$ to a uniform distribution over the number $K$ of available components and do not further optimize this distribution. This is a useful definition to increase the entropy of the mixture model.

**Parameterization of the context distribution** $\pi(\mathbf{c})$. Due to the relation $\pi(\mathbf{c}) = \sum_o \pi(\mathbf{c}|o)\pi(o)$, $\pi(\mathbf{c})$ is defined by $\pi(\mathbf{c}|o)$ and does not need explicit modelling.

**Parameterization of the gating distribution** $\pi(o|\mathbf{c})$. Due to the relation $\pi(o|\mathbf{c}) = \frac{\pi(\mathbf{c}|o)\pi(o)}{\pi(\mathbf{c})}$ we do not need an explicit parameterization of $\pi(o|\mathbf{c})$ and can easily calculate the probabilities for choosing the expert $o$ given a context $\mathbf{c}$.

# B   Using Motion Primitives in the Context of Reinforcement Learning

Motion Primitives (MPs) are a low-dimensional representation of a trajectory. For instance, instead of parameterizing a desired joint-level trajectory as the single state in each time step, MPs introduce a low-dimensional parameter vector $\boldsymbol{\theta}$ which concisely defines the trajectory to follow. The generation of the trajectory depends on the method that is used. Probabilistic Movement Primitives (ProMPs) Paraschos et al. (2013) for example define the desired trajectory as a simple linear function $\boldsymbol{\tau} = \boldsymbol{\Phi}^T \boldsymbol{\theta}$, where $\boldsymbol{\Phi}$ are time-dependent basis functions (e.g. normalized radial basis functions). Dynamic Movement Primitives (DMPs) Schaal (2006) rely on a second order dynamic system which provides smooth trajectories in the position and velocity space. Recently Probabilistic Dynamic Movement Primitives (ProDMPs) were introduced by Li et al. (2023) and combines the advantages of both methods, that is the easy generation of trajectories and smooth trajectories. We therefore rely on ProDMPs througout this work.

In the context of reinforcement learning, the policy $\pi(\boldsymbol{\theta}|\mathbf{c})$, or in our case an expert $\pi(\boldsymbol{\theta}|\mathbf{c}, o)$ defines a distribution over the paremeters $\boldsymbol{\theta}$ of the MP depending on the observed context $\mathbf{c}$. This allows the policy to quickly adapt to new tasks defined by $\mathbf{c}$.

# C   Experiment Details

## C.1   Hyperparameters and Environment Details

### C.1.1   Ablation Studies

**Environment.** We use the same table tennis environment as presented in (Otto et al., 2023), in which a 7 Degree of Freedom (DoF) robot has to return a ball to a desired ball landing position. The **context** is the four dimensional space of the ball's initial landing position ( $x \in [-1, -0.2]$, $y \in [-0.65, 0.65]$) on the robot's table side and the desired ball landing position ($x \in [-1.0, -0.2]$, $y \in [-0.6, 0.6]$) on the opponent's table side. The robot is controlled with torques on joint-level in each time-step. The torques are generated by the tracking controller (PD-controller) that tracks

the desired trajectory generated by the motion primitive. We consider three basis functions per joint resulting in a 21 dimensional parameter ($\boldsymbol{\theta}$) space. We additionally allow the agent to learn the trajectory length and the starting point of the trajectory. Note that the starting point allows the agent to define when after the episode's start the generated desired trajectory should be tracked. Induced by the varying contexts, this is helpful to react to the varying time the served ball needs to reach a positional space that is convenient to hit the ball with the robot's racket. Overall the **parameter space** is 23 dimensional. The task is considered successful if the returned ball lands on the opponent's side of table and within $\leq 0.2$m to the goal location.

The **reward function** is unchanged from (Otto et al., 2023) and is defined as

$$
R_{task} = \begin{cases}
0, & \text{if cond. 1} \\
0.2 - 0.2\tanh\left(\min\|\mathbf{p}_r - \mathbf{p}_b\|^2\right), & \text{if cond. 2} \\
3 - 2\tanh\left(\min\|\mathbf{p}_r - \mathbf{p}_b\|^2\right) - \tanh\left(\|\mathbf{p}_l - \mathbf{p}_{goal}\|^2\right), & \text{if cond. 3} \\
6 - 2\tanh\left(\min\|\mathbf{p}_r - \mathbf{p}_b\|^2\right) - 4\tanh\left(\|\mathbf{p}_l - \mathbf{p}_{goal}\|^2\right), & \text{if cond. 4} \\
7 - 2\tanh\left(\min\|\mathbf{p}_r - \mathbf{p}_b\|^2\right) - 4\tanh\left(\|\mathbf{p}_l - \mathbf{p}_{goal}\|^2\right), & \text{if cond. 5}
\end{cases}
$$

where $\mathbf{p}_r$ is the executed trajectory position of the racket center, $\mathbf{p}_b$ is the executed position trajectory of the ball, $\mathbf{p}_l$ is the ball landing position, $\mathbf{p}_{goal}$ is the target position. The different conditions are

- cond. 1: the end of episode is not reached,
- cond. 2: the end of episode is reached,
- cond. 3: cond.2 is satisfied and robot did hit the ball,
- cond. 4: cond.3 is satisfied and the returned ball landed on the table,
- cond. 5: cond.4 is satisfied and the landing position is at the opponent's side.

The episode ends when any of the following conditions are met

- the maximum horizon length is reached
- ball did land on the floor without hitting
- ball did land on the floor or table after hitting

The whole desired trajectory is obtained ahead of environment interaction, making use of this property we can collect some samples without physical simulation. The reward function based on this desired trajectory is defined as

$$
r_{traj} = -\sum_{(i,j)} |\tau_{ij}^d| - |q_j^b|, \quad (i,j) \in \{(i,j) \mid |\tau_{ij}^d| > |q_j^b|\}
$$

where $\tau^d$ is the desired trajectory, $i$ is the time index, $j$ is the joint index, $q^b$ is the joint position upper bound. The desired trajectory is considered as invalid if $r_{traj} < 0$, an invalid trajectory will not be executed on the robot. The overall reward is defined as:

$$
r = \begin{cases}
r_{traj}, & r_{traj} < 0 \\
r_{task}, & \text{otherwise}
\end{cases}
$$

**SVSL.** SVSL requires designing a guiding punishment term for context samples that are not in a valid region. For the four dimensional context space in table tennis this can be easily done using quadratic functions (as proposed in the original work (Celik et al., 2022)):

$$
R_c(\mathbf{c}) = -20 \cdot d_c^2,
$$

where $d_c^2$ is the distance of the current context $\mathbf{c}$ to the valid context region.

**SVSL Hyperparameters** All hyperparameters are summarized in the Table 1.

**Di-SkilL and BBRL Hyperparameters** All hyperparameters are summarized in the Table 2.

| | |
|---|---|
| add component every iteration | 1000 |
| fine tune all components every iteration | 50 |
| number component adds | 1 |
| number initial components | 1 |
| number total components | 20 |
| number traj. samples per component per iteration | 200 |
| $\alpha$ | 0.0001 |
| $\beta$ | 0.5 |
| expert KL-bound | 0.01 |
| context KL-bound | 0.01 |

Table 1: Hyperparameters for SVSL

| | Di-SkilL | BBRL |
|---|---|---|
| critic activation | tanh | tanh |
| hidden sizes critic | [8,8] | [32, 32] |
| initialization | orthogonal | orthogonal |
| lr critic | 0.0003 | 0.0003 |
| optimizer critic | adam | adam |
| ciritc epochs | 100 | 100 |
| activation context distribution | tanh | – |
| epochs context distribution | 100 | – |
| hidden sizes context distr | [16,16] | – |
| initialization | orthogonal | – |
| lr context distribution | 0.0001 | – |
| optimizer context distr | adam | – |
| batch size per component | 50 | 209 |
| number samples from environment distribution | 5000 | – |
| number samples per component | 50 | 209 |
| normalize advantages | True | True |
| expert activateion | tanh | tanh |
| epochs | 100 | 100 |
| hidden sizes expert | [64] | [32] |
| lr policy | 0.0003 | 0.0003 |
| covariance type | full | full |
| alpha | 0.001 | – |
| beta | 0.5 | – |
| number components | 5 | – |
| covariance bound | 0.005 | 0.001 |
| mean bound | 0.05 | 0.05 |
| projection type | KL | KL |
| trust region coefficient | 100 | 25 |

Table 2: Hyperparameters for Di-SkilL and BBRL for the ablations.

### C.1.2 EXTENDED TABLE TENNIS TASK.

**Environment.** We extend the table tennis environment described in Appendix C.1.1 by additionally including the ball's initial velocity into the context space. We define the initial velocity $v_x \in \left[1.5\frac{m}{s}, 4\frac{m}{s}\right]$. Note that every single constellation within the resulting context space is a valid context. There exist ball landing positions which can not be set along with a subset of the initial velocity range. This makes designing a guiding punishment term for SVSL especially difficult. We adopt the **parameter space** and the **reward function** as defined in the standard table tennis environment as described in Appendix C.1.1.

**Di-SkilL and BBRL Hyperparameters** All hyperparameters are summarized in the Table 3.

| | Di-SkilL | BBRL |
|---|---|---|
| critic activation | tanh | tanh |
| hidden sizes critic | [8,8] | [32, 32] |
| initialization | orthogonal | orthogonal |
| lr critic | 0.0003 | 0.0003 |
| optimizer critic | adam | adam |
| ciritc epochs | 100 | 100 |
| activation context distribution | tanh | – |
| epochs context distribution | 100 | – |
| hidden sizes context distr | [16,16] | – |
| initialization | orthogonal | – |
| lr context distribution | 0.0001 | – |
| optimizer context distr | adam | – |
| batch size per component | 50 | 209 |
| number samples from environment distribution | 5000 | – |
| number samples per component | 50 | 209 |
| normalize advantages | True | True |
| expert activateion | tanh | tanh |
| epochs | 100 | 100 |
| hidden sizes expert | [128] | [32,32] |
| lr policy | 0.0003 | 0.0003 |
| covariance type | full | full |
| alpha | 0.001 | – |
| beta | 0.5 | – |
| number components | 10 | – |
| covariance bound | 0.005 | 0.0005 |
| mean bound | 0.05 | 0.05 |
| projection type | KL | KL |
| trust region coefficient | 100 | 25 |

Table 3: Hyperparameters for Di-SkilL and BBRL for the extended Table Tennis Task.

### C.1.3 EXTENDED BOX PUSHING TASK

**Environment.** We adapt the box pushing environment as presented in (Otto et al., 2023), by changing major parts of the context space. The goal of the box pushing task is to move a box to a specified goal location and orientation using the seven DoF Franka Emika Panda. The newly **context space** (compared to the original version in (Otto et al., 2023)) are described in the follown. We increase the box' goal position range to $x_g \in [0.3, 0.6], y_g \in [-0.7, 0.45]$, and keep the goal orientation angle $\phi \in [0 rad, 2\pi rad]$. Additinally, we include an obstacle in between the initial box and the box' goal. The range of the obstacles position is $x_o \in [0.3, 0.6], y_o \in [-0.3, 0.15]$. note that we guarantee a distance of at least 0.15m between the obstacle's position and the initial position as well as at least 0.15m between the obstacle's position and the box' goal position.

The robot is controlled via torques on joint level. We use four basis functions per DoF, resulting in a **parameter space** of 28 dimensions. We consider an episode successful if the box' orientation around the z-axis error is smaller than 0.5 rad and the position error is smaller than 0.05m.

The **sparse-in-time reward function** is up to a scaling parameter the same as presented in (Otto et al., 2023). We describe the whole reward function in the following.

The box' distance to the goal position is

$$R_{\text{goal}} = \|\mathbf{p} - \mathbf{p}_{goal}\|,$$

where $\mathbf{p}$ is the box position and $\mathbf{p}_{goal}$ is the goal position. The rotation distance is defined as

$$R_{\text{rotation}} = \frac{1}{\pi} \arccos |\mathbf{r} \cdot \mathbf{r}_{goal}|,$$

where $\mathbf{r}$ and $\mathbf{r}_{goal}$ are the box orientation and goal orientation quaternion respectively. The incentive to keep the rod within the box is defined as

$$R_{\text{rod}} = \text{clip}(||\mathbf{p} - \mathbf{h}_{pos}||, 0.05, 10),$$

where $\mathbf{h}_{pos}$ is the position of the rod tip. Similarly, to incentivize to maintain the rod in a desired rotation, the reward

$$R_{\text{rod\_rotation}} = \text{clip}(\frac{2}{\pi} \arccos |\mathbf{h}_{rot} \cdot \mathbf{h}_0|, 0.25, 2)$$

is defined, where $\mathbf{h}_{rot}$ and $\mathbf{h}_0 = (0, 1, 0, 0)$ are the current and desired rod orientation in quaternion respectively. To incentivize the robot to stay within the joint and velocity bounds, the error

$$\text{err}(\mathbf{q}, \dot{\mathbf{q}}) = \sum_{i \in \{i | |q_i| > |q_i^b|\}} (|q_i| - |q_i^b|) + \sum_{j \in \{j | |\dot{q}_j| > |\dot{q}_j^b|\}} (|\dot{q}_j| - |\dot{q}_j^b|)$$

is used, where $\mathbf{q}$, $\dot{\mathbf{q}}$, $\mathbf{q}^b$, and $\dot{\mathbf{q}}^b$ are the robot's joint positions and velocities as well as their respective bounds. To learn low-energy motions, the per-time action (torque) cost

$$\tau_t = \sum_i^K (a_t^i)^2,$$

is used. The resulting temporal sparse reward is given as

$$R_{\text{tot}} = \begin{cases} -R_{\text{rod}} - R_{\text{rod\_rotation}} - 0.02\tau_t - \text{err}(\mathbf{q}, \dot{\mathbf{q}}), & t < T, \\ -R_{\text{rod}} - R_{\text{rod\_rotation}} - 0.02\tau_t - \text{err}(\mathbf{q}, \dot{\mathbf{q}}) - 350R_{\text{goal}} - 200R_{\text{rotation}}, & t = T, \end{cases}$$

where $T = 100$ is the horizon of the episode. The reward gives relevant information to solve the ask only in the last time step of the episode, which makes exploration hard.

**Di-SkilL and BBRL Hyperparameters** All hyperparameters are summarized in the Table 4.

|  | Di-SkilL | BBRL |
|---|---|---|
| critic activation | tanh | tanh |
| hidden sizes critic | [32,32] | [32, 32] |
| initialization | orthogonal | orthogonal |
| lr critic | 0.0003 | 0.0003 |
| optimizer critic | adam | adam |
| ciritc epochs | 100 | 100 |
| activation context distribution | tanh | – |
| epochs context distribution | 100 | – |
| hidden sizes context distr | [16,16] | – |
| initialization | orthogonal | – |
| lr context distribution | 0.0001 | – |
| optimizer context distr | adam | – |
| batch size per component | 50 | 399 |
| number samples from environment distribution | 5000 | – |
| number samples per component | 50 | 399 |
| normalize advantages | True | True |
| expert activateion | tanh | tanh |
| epochs | 100 | 100 |
| hidden sizes expert | [32,32] | [64,64] |
| lr policy | 0.0003 | 0.0003 |
| covariance type | diagonal | diagonal |
| alpha | 0.01 | – |
| beta | 64 | – |
| number components | 10 | – |
| covariance bound | 0.001 | 0.005 |
| mean bound | 0.05 | 0.05 |
| projection type | KL | KL |
| trust region coefficient | 25 | 25 |

Table 4: Hyperparameters for Di-SkilL and BBRL for the extended Box Pushing task.

## C.2 EXTENDED 5-LINK REACHER TASK

**Environment.** In the 5-Link Reacher task, a 5-link planar robot has to reach a goal position with its tip. The reacher's initial position is straight to the right. This task is difficult to solve especially for episodic RL methods, as it introduces multi-modality in the behavior space. (Otto et al., 2023) avoided this multi-modality by constraining the y coordinate of the goal position to $y \geq 0$, i.e. the first two quardants. We adapt the 5Link-Reacher task by increasing the context space to the full space, i.e. all four quadrants. We consider 5 basis functions per joint leading to a 25 dimensional **parameter space**. We consider the **sparse reward function** presented in (Otto et al., 2023) as

$$R_{\text{tot}} = \begin{cases} -\tau_t & t < T, \\ -\tau_t - 200 R_{\text{goal}} - 10 R_{\text{vel}} & t = T, \end{cases}$$

where

$$R_{\text{goal}} = \|\mathbf{p} - \mathbf{p}_{goal}\|_2$$

and

$$\tau_t = \sum_i^K (a_t^i)^2,$$

. The sparse reward only returns the task reward in the last time step T and additionally adds a velocity penalty $R_{\text{vel}} = \sum_i^K (\dot{q}_T^i)^2$. The joint velocities are denoted as $\dot{\mathbf{q}}$. This velocity penalty avoids overshooting in the last time step.

**Di-SkilL and BBRL Hyperparameters** All hyperparameters are summarized in the Table 5.

| | Di-SkilL | BBRL |
|---|---|---|
| critic activation | tanh | tanh |
| hidden sizes critic | [32,32] | [32, 32] |
| initialization | orthogonal | orthogonal |
| lr critic | 0.0003 | 0.0003 |
| optimizer critic | adam | adam |
| ciritc epochs | 100 | 100 |
| activation context distribution | tanh | – |
| epochs context distribution | 100 | – |
| hidden sizes context distr | [16,16] | – |
| initialization | orthogonal | – |
| lr context distribution | 0.0001 | – |
| optimizer context distr | adam | – |
| batch size per component | 50 | 240 |
| number samples from environment distribution | 5000 | – |
| number samples per component | 50 | 240 |
| normalize advantages | True | True |
| expert activateion | tanh | tanh |
| epochs | 100 | 100 |
| hidden sizes expert | [32,32] | [64,64] |
| lr policy | 0.0003 | 0.0003 |
| covariance type | full | full |
| alpha | 0.0001 | – |
| beta | 16 | – |
| number components | 10 | – |
| covariance bound | 0.001 | 0.005 |
| mean bound | 0.05 | 0.05 |
| projection type | KL | KL |
| trust region coefficient | 100 | 25 |

Table 5: Hyperparameters for Di-SkilL and BBRL for the extended 5-Link Reacher task.

## C.3 ROBOT MINI GOLF TASK

**Environment.** In the robot mini golf task the agent needs to hit a ball while avoiding the two obstacles, such that it passes the tight goal to achieve a bonus. The **context space** consists of the ball's initial x-position $x_{ball} \in [0.25m, 0.6m]$, the XY positions of the green obstacle $x_{obs} \in [0.3, 0.6]$ and $y_{obs} \in [-0.5, -0.1]$ and the x positions of the goal $x_{ball} \in [0.25, 0.6]$. The **parameter space** is 29 dimensional resulting from the 4 basis functions per joint and an additional duration parameter which allows the robot to learn the duration of the trajectory. The robot starts always at the same position. The **reward** function consists of three stages:

$$R_{task} = \begin{cases} -0.0005 \cdot \tau_t, & \text{if cond. 1} \\ 0.2 - 0.2 \tanh\left(\min\|\mathbf{p}_r - \mathbf{p}_b\|\right), & \text{if cond. 2} \\ 2 - 2\tanh\left(\min\|\mathbf{p}_b - \mathbf{p}_g\|\right) - \tanh\left(\|\mathbf{p}_{b,y} - \mathbf{p}_{thresh,y}\|\right), & \text{if cond. 3} \\ 6, \end{cases}$$

where the individual conditions are

- cond. 1: the end of episode is not reached,
- cond. 2: the end of episode is reached and robot did not hit the ball,
- cond. 3: the end of episode is reached and robot has hit the ball, but ball didn't pass the goal
- cond. 4: the end of episode is reached, robot has hit the ball and the ball has passed the goal for at least 0.75m

The episode ends when the maximum horizon length $T = 100$ is achieved. We again make use of the advantage that we obtain the whole desired trajectory ahead of the environment interaction, such

that we can collect some samples without physical simulation. The reward function based on this desired trajectory is defined as

$$r_{traj} = \sum_{(i,j)} |\tau_{ij}^d| - |q_j^b|, \quad (i,j) \in \{(i,j) \mid |\tau_{ij}^d| > |q_j^b|\}$$

where $\tau^d$ is the desired trajectory, $i$ is the time index, $j$ is the joint index, $q^b$ is the joint position upper bound. The desired trajectory is considered as invalid if $r_{traj} < 0$, an invalid trajectory will not be executed on the robot. Additionally, we provide a punishment, if the agent samples invalid duration times

$$r_{dur} = -3 \left( \max(0, t_d - td, max) + \max(0, t_{d,mint} - t_d) \right),$$

where $t_{d,max} = 1.7s, t_{d,min} = 0.45s$ and $t_d$ is the duration in seconds chosen by the agent. The overall reward is defined as:

$$r = \begin{cases} r_{traj}, & -20(r_{traj} + r_{dur}) - 5 \quad \text{if invalid duration, or trajectory} \\ r_{task}, & \text{otherwise} \end{cases}$$

**Di-SkilL and BBRL Hyperparameters**  All hyperparameters are summarized in the Table 6.

|  | Di-SkilL | BBRL |
|---|---|---|
| critic activation | tanh | tanh |
| hidden sizes critic | [32,32] | [32, 32] |
| initialization | orthogonal | orthogonal |
| lr critic | 0.0003 | 0.0003 |
| optimizer critic | adam | adam |
| ciritc epochs | 100 | 100 |
| activation context distribution | tanh | – |
| epochs context distribution | 100 | – |
| hidden sizes context distr | [16,16] | – |
| initialization | orthogonal | – |
| lr context distribution | 0.0001 | – |
| optimizer context distr | adam | – |
| batch size per component | 50 | 500 |
| number samples from environment distribution | 5000 | – |
| number samples per component | 50 | 500 |
| normalize advantages | True | True |
| expert activateion | tanh | tanh |
| epochs | 100 | 100 |
| hidden sizes expert | [64,64] | [128,128] |
| lr policy | 0.0003 | 0.0003 |
| covariance type | full | full |
| alpha | 0.0001 | – |
| beta | 2 | – |
| number components | 10 | – |
| covariance bound | 0.005 | 0.001 |
| mean bound | 0.05 | 0.05 |
| projection type | KL | KL |
| trust region coefficient | 100 | 25 |

Table 6: Hyperparameters for Di-SkilL and BBRL for the mini golf task.

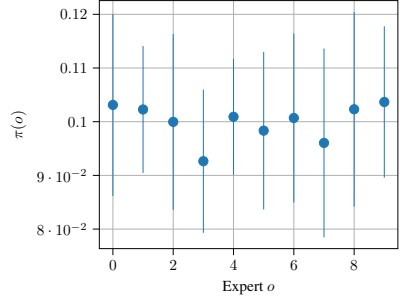

(a) $\pi(o)$ for the extended table tennis task.

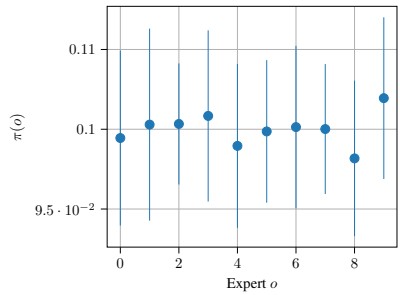

(b) $\pi(o)$ for the extended box pushing task.

Figure 6: **Marginal distributions ($\pi(o)\pm std$) on the extended a) table tennis and b) box pushing task.** We calculate $\approx(o)\frac{1}{N}\sum_i \pi(o|\mathbf{c}_i)$ for $N=1e6$ context samples drawn from the environment's context distribution $p(\mathbf{c})$. The probabilities are approximately uniformly distributed over the number of experts, which shows that all of them are used by the mixture of experts model.

# D  ANALYZING THE MIXTURE OF EXPERTS MODEL TRAINED WITH DI-SKILL

In this section we are going to analyze the learned mixture models w.r.t. the usage of the individual experts. A good indicator, whether the experts are effectively used is to calculate the marginal probability of the gating distribution $\pi(o|\mathbf{c})$. We calculate this entity by marginalizing out the context $\int_{\mathbf{c}} \pi(o|\mathbf{c})p(\mathbf{c})$. We sample N = 1 million contexts from the environment' context distribution $p(\mathbf{c})$ leading to the monte carlo estimate $\pi(o) \approx \frac{1}{N}\sum_i \pi(o|\mathbf{c}_i)$. We repeat this calculation for all 24 seeds in each environment and report the mean and the standard deviation of the probabilities for each expert.

Additionally, to analyze the usage of the experts on the context level, we use the N context samples to calculate how many contexts have at least two experts with higher probability than 5%. Given that we use 10 experts in our experiments this is a good threshold. Please note that it is difficult to determine whether there exist several solutions/skills for every context as this highly depends on the task. For example, for the table tennis agent, most experts might be invalid, if the initial ball velocity is very small, as the experts that hit the ball at the height of the robot itself won't be able to hit those as they require completely different motions. Similarly, for the box pushing task several solutions, e.g. going right and left of the obstacle, might not be feasible because the obstacle is not far enough from the table's edge, or the robot itself such that the box can not be dragged to the goal. Yet, to obtain a measurement of the diversity, we report the aforementioned metrics.

The calculated marginal distributions can be seen in Fig. 6. For both environments we obtain approximately uniform distributions indicating that we all experts are used. Additionally, for the extended table tennis task, $42\% \pm 4.5\%$ of the contexts have at least two experts that have a higher probability than 5% for $\pi(o|\mathbf{c})$. For the extended box ushing task, For the TT task, $42\% \pm 4.5\%$ of the contexts have at least two experts that have a higher probability than 5% for $\pi(o|\mathbf{c})$. The marginal gating distribution is calculated to $60\% \pm 5\%$ of the contexts have at least two experts that have a higher probability than 5% for $\pi(o|\mathbf{c})$.

