# OpenReview forum: "Reinforcement Learning of Diverse Skills using Mixture of Deep Experts"
_ICLR.cc/2024/Conference — Submitted to ICLR 2024_

### Official Review · Reviewer_ydwi · 2023-10-28

**Soundness:** 2 fair
**Presentation:** 1 poor
**Contribution:** 2 fair
**Rating:** 5
**Confidence:** 3

**Summary:**

The paper proposes a method for learning diverse skills to solve different contexts of the same task. The method is designed to prioritize experts that are promising in different contexts. The algorithm involves training each experts in the corresponding task context and updating the joint distribution of experts and task contexts. Experimental findings indicate that this approach effectively trains experts in two robotics domains and yields a certain degree of diversity among the trained experts.

**Strengths:**

The idea is interesting and could has the potential be applied in more complex domains.

**Weaknesses:**

1. The motivation behind the research is not clearly articulated. It is unclear whether the authors intend to discover diverse solutions within the same task or seek experts for all tasks/contexts.
2. The paper lacks sufficient detail regarding the definition of the mixture of experts model, including the definition of an expert. Furthermore, the relationships between context (c), expert (o), and the parameter θ are not adequately explained.
3. The experimental section appears to be confined to relatively simple scenarios, and the demonstrated diversity of the trained experts is limited.

**Questions:**

1. What is the goal of the method? Is it trying to discover diverse solutions or seek experts for different contexts/tasks?
2. What is the definition of a expert and how it is executed in certain context/task.

---

> ### Author Response · Authors · 2023-11-17
> **Reply to Reviewer ydwi**
>
> Thank you very much for your insightful comments and the time you have dedicated to reviewing our manuscript. We appreciate the opportunity to clarify the points that might have seemed ambiguous, which may have led to certain misunderstandings regarding our work. We updated the paper and marked the changed text in blue. Before the end of the author-review discussion deadline, we will address the concerns of the other reviewers including additional experiments.

---

> > ### Author Response · Authors · 2023-11-17
> > **Reply to Reviewer ydwi**
> >
> > *"The motivation behind the research is not clearly articulated. It is unclear whether the authors intend to discover diverse solutions within the same task or seek experts for all tasks/contexts."* Also related to the Question *"What is the goal of the method? Is it trying to discover diverse solutions or seek experts for different contexts/tasks?"*
> >
> > * We acknowledge your concerns regarding the unclear motivation of our work. It appears there might have been a misinterpretation of our approach/intentions. To clarify,  Di-SkilL is a reinforcement learning method to train a Mixture of Experts (MoE) model that is capable of learning both: a) solutions for all contexts and b) diverse solutions within the same task defined by a specific context. To enhance the clarity of our paper, we have revised the relevant parts to ensure this concept is more clearly communicated. Moreover, in the following paragraphs, we summarize the main points:
> >     * Contextual Episodic Policy Search (CEPS) generally considers training an agent that generalizes to (continuous-valued) contexts and a  context defines the task within an environment. In the robot table tennis environment, for example, a context can be the ball’s initial velocity, ball landing position, and desired ball landing position on the opponent’s side. CEPS assumes that the environment has its internal context distribution $p(c)$, which is unknown to the agent. During training, the agent observes a context $c$ and proposes a motion primitive parameter $\theta$ which results in a desired trajectory. This is formalized in the optimization problem in Eq. (1).
> >     The goal of Di-SkilL is to train a mixture of experts (MoE) policy that generalizes well to all contexts that are within the support of      $p(c)$. At the same time, Di-SkilL learns different solutions, i.e. behaviors to a specific context $c$, resulting in different strategies for the task defined by $c$. If we assume that the context in table tennis defines the initial ball velocity, ball landing position, and desired ball landing position on the opponent’s side, Di-SkilL is able to learn different striking types in the same context $c$. We tried to visualize these behaviors in Fig. 4 and in the videos provided in the supplementary material.
> >
> > *"The paper lacks sufficient detail regarding the definition of the mixture of experts model, including the definition of an expert. Furthermore, the relationships between context (c), expert (o), and the parameter θ are not adequately explained."* Also related to Question *"What is the definition of a expert and how it is executed in certain context/task."*
> > * The general definition of a mixture of experts model is provided in Eq. 2. However, this definition requires parameterizing a gating distribution $\pi(o|c)$, which allows sampling an expert $o$  after observing a (continuous-valued) context $c$. Due to the finite number of experts, $\pi(o|c)$ can e.g. be a conditioned categorical distribution. The expert $\pi(\theta|c,o)$ is a parameterized distribution (usually a Gaussian) that maps the contexts $c$ to motion primitive parameters $\theta$.
> > However, this form of parameterization does not allow for automatic curriculum learning. Therefore, we rely on the MoE definition in Eq. 3. This definition was already used in prior works (citations were added to the paper). In our work, the per-expert context distribution is parameterized to provide the possibility to adjust its curriculum for each expert $o$. As suggested by the $\sum_o$ in Eq. 2 and Eq. 3, there is a discrete number of experts in each MoE model.
> >
> > * An expert represents a mapping of contexts $c$ to motion primitive parameters $\theta$. We describe this in Section 2 (CEPS). In our work, we parameterize each expert with a neural network, where the mean is a context-dependent function and the covariance can either be context-dependent or not. We tried to visualize this relation in Fig. 1 and further mentioned it in Section 3.3 “Expert Update”. We extended Section 2 where we explain the parameterization of each expert and added a dedicated section in Appendix A where we describe the parameterizations of each part of the MoE model in detail.

---

> > > ### Author Response · Authors · 2023-11-17
> > > **Reply to Reviewer ydwi continued**
> > >
> > > * The relationships between context $c$, expert $o$, and the parameter $\theta$ are formalized by the mixture model as a whole. During inference, we observe a context $c$, and choose an expert $o$ by sampling from the gating $\pi(o|c)$. Note that the gating distribution is defined as $\pi(o|c)=\frac{\pi(c|o)\pi(o)}{\pi(c)}$(transition from Eq. (2) to Eq. (3)) and results in a categorical distribution over the number of experts for each context $c$.
> > > The expert $\pi(\theta|c,o)$ is a Gaussian distribution parameterized by a neural network and allows sampling motion primitive parameters $\theta$. During training, the agent does not need to choose an expert based on an observed context $c$, but rather can set a context in the environment by sampling it from its context distribution $\pi(c|o)$. Afterward, the expert samples a motion primitive parameter $\theta$. We visualized these relationships in Fig. 1. The probabilistic graphical models in Fig. 2 a) (during inference) and in Fig. 2 b) (during training) additionally show the relations of the random variables. To make these relations more clear, we have extended the text describing the Mixture of Experts policy in Section 2 and additionally provide more details on the definition of the MoE in Appendix A. Furthermore, we provide a detailed description of the motion primitives in the context of reinforcement learning in Appendix B. If the reviewer has further suggestions to improve the paper, we are happy to include them.
> > >
> > > *"The experimental section appears to be confined to relatively simple scenarios, and the demonstrated diversity of the trained experts is limited."*
> > >
> > > * We agree that evaluating our method on more tasks would strengthen the work. We are currently working on further experiments and plan to provide them before the end of the author-reviewer discussion.
> > > * Additionally, we would like to emphasize that the “table tennis” and the “box pushing with obstacles” environments are extensions of the environments provided in [1], where they were extensively tested. Moreover, we extend the environments by increasing the dimensionality of the context spaces: for the table tennis task, we not only vary the initial ball landing position and the ball goal landing position but also vary the ball’s initial velocity. For the box-pushing environment, we additionally include a varying obstacle location. Both environments are challenging benchmark tasks using the state-of-the-art physics engine Mujoco[2]. This makes exploration even more challenging because the agent needs to infer the dynamics solely based on environmental interactions. In the box-pushing task, for example, friction leads to highly non-linear behaviors and for the table tennis task, the understanding of the ball’s trajectory is more difficult due to the varying initial velocity.  Additionally, successfully completing an episode is challenging due to the strict definitions as described in Section 4. This means that the robot has to perform very accurate and precise motions to achieve a position and rotation error below the success-threshold.
> > >
> > > [1] Otto et al, Deep black-box reinforcement learning with movement primitives.
> > >
> > >
> > > [2] Todorov et al, MuJoCo: A physics engine for model-based control

---

> > ### Comment · Reviewer_ydwi · 2023-11-21
> > **Reply to the Authors**
> >
> > Appreciate the comprehensive responses. The authors have effectively addressed the majority of my questions about the definition of the model. But still I have concerns regarding the constrained evaluation. I've decided to increase my score.

---

> > > ### Author Response · Authors · 2023-11-22
> > > **RE: Reply to the Authors**
> > >
> > > We thank the reviewer for the answer and for raising the score. To address the reviewer's remaining concerns, we have added a dedicated analysis of the learned model in Appendix D. Additionally we want to mention that we have added additional baselines 'LinDi-SkilL' as an alternative to SVSL on all environments (please see the answers to Reviewer w2WU). We are working on additional experiments and plan to provide them before the end of the discussion period.

---

> > > > ### Author Response · Authors · 2023-11-22
> > > > **RE: Reply to the Authors**
> > > >
> > > > To clarify the remaining concerns of the reviewer, we would like to inform you that we have
> > > >
> > > > * added a comparison to PPO in Fig. 4 b)
> > > > * added a 5-Link Reacher benchmark with sparse rewards (Fig. 3a) + Fig. 3c)
> > > >  * added a challenging robot mini golf benchmark with non-markovian rewards (Fig. 3a) + Fig. 4c)
> > > >
> > > > If there are any further questions regarding our work, we would be happy to provide additional clarification for the reviewer

---

> ### Comment · Area_Chair_UsJw · 2023-11-20
> **Author-Reviewer Discussion Period Ending November 22**
>
> Hi,
>
> Thanks for your help with the review process!
>
> There are only two days remaining for the author-reviewer discussion (November 22nd). Please read through the authors' response to your review and comment on the extent to which it addresses your questions/concerns.
>
> Best,
> AC

---

### Official Review · Reviewer_w2WU · 2023-10-28

**Soundness:** 3 good
**Presentation:** 2 fair
**Contribution:** 2 fair
**Rating:** 5
**Confidence:** 3

**Summary:**

This paper investigates the problem of learning diverse skills in contextual RL problems. It achieves so in the framework of contextual episode-based policy search and aims to learn a mixture of expert policies. It follows the previous work SVSL [Celik et al. 2022] to jointly learn a per-expert context curriculum $\pi(c|o)$ and a context conditioned policy $\pi(\theta|c, o)$. The key contributions of this work is (1) using softmax-based per-expert context distribution to model the curriculum which enables validity and multi-modality of the sampled context curriculum; (2) using trust-region and PPO to stabilize the bi-level policy training. The proposed approach is compared against two baselines BBRL and SVSL on Table Tennis Env and Box Pushing Env and shown to outperform baselines.

**Strengths:**

1. The topic of optimizing a set of expert policies with diverse strategies for the same task is beneficial to improving robustness of the robotic control and helps capture the multi-modality nature of some real-world tasks.

2. The idea of achieving automatic multi-modality context curriculum learning via applying softmax on sampled context is intuitive.

3. The experiments show the proposed algorithm performs better or at least similar to baseline algorithms on evaluated problems.

**Weaknesses:**

1. While the concept of the proposed technique is easy to follow, some important details are missing and it might affect the reproducibility of the proposed approach.

3. The novelty over previous work seems incremental.

3. More extensive evaluation are needed.

**Questions:**

1. To better understand the action space of the contextual episode-based policy, could the author give some details or examples of the concept of motion primitives and how to convert the policy parameters into the episode-wise robot action trajectory?

2. Eq (3) seems to be not original from this work, a proper reference would help readers to understand the background of this line of work.
The derivation from Eq (4) to Eq (5) and Eq (6) is unclear. It would be more clear to have an intermediate objective which is jointly optimizing for $\pi(\theta|c, o)$ and $\pi(c|o)$, and derive from there to have two separate objectives for bi-level optimization.

3. In Section 3.1, it says “mapping the context $c$ to a mean vector and a covariance matrix” and “Note that in most cases a context dependent covariance matrix is not useful and introduces unnecessary complex relations.” It is confusing that whether the covariance matrix in the implementation is context dependent.

4. Line 10 in Section 3.2, should “Fig. 2c” be “Fig. 2d”?

5. Which terms in Eq (8) and Eq (9) accounts for encouraging the coverage of the context space by experts? From the formulation, it seems to try to learn a set of policy each of which can solve the entire task space as much as possible. The learning of policies seem to be relatively independent and is it possible to learn a set of experts whose preferred context distributions are the same.

6. More testing environment description would be helpful. Some details about action space and reward definitions are missing for both tasks.

7. Evaluating the algorithm on more environments will make the comparison more thorough. For example, it would be helpful to evaluate on the other tasks used in Otto et al. 2023.

8. It would also helpful to show complete comparison against both SVSL and BBRL on all evaluated tasks (at least provide comparison plots in Appendix)

9. Given the proposed approach is built upon SVSL with two improvements, it would be great to do ablation study on both improvement techniques.

10. The multi-modality in this work is achieved by mixture of experts, however each expert is still modeled by uni-model gaussian policy.

11. Recent work (Huang et al. 2023 [1]) proposes some multi-modal policy parameterization. How is the proposed approach compare to this work and can the proposed approach enhanced by the policy reparameterization from [1]?

12. Is this proposed approach also applicable to step-based RL problems?

[1] Huang et al, Reparameterized Policy Learning for Multimodal Trajectory Optimization.

---

> ### Comment · Area_Chair_UsJw · 2023-11-20
> **Author-Reviewer Discussion Period Ending November 22**
>
> Hi,
>
> Thanks for your help with the review process!
>
> There are only two days remaining for the author-reviewer discussion (November 22nd). Please read through the authors' response to your review and comment on the extent to which it addresses your questions/concerns.
>
> Best,\
> AC

---

> ### Author Response · Authors · 2023-11-20
> **Reply to Reviewer w2WU**
>
> We would like to thank the reviewer for the insightful comments and the dedicated time invested to review our work. We would like to clarify the reviewer’s concerns and questions in the following. We have updated the paper and marked the changed text motivated by suggestions of the reviewer in green.
>
>
>
> *"While the concept of the proposed technique is easy to follow, some important details are missing and it might affect the reproducibility of the proposed approach".*
>
> * We presume that the reviewer is referring to the points in the “Questions” part and kindly ask to specify what are missing details apart from the questions if there are any.
> * Additionally, we want to note that we will be open-sourcing the code and the environments upon acceptance
>
> 1. *"To better understand the action space of the contextual episode-based policy, could the author give some details or examples of the concept of motion primitives and how to convert the policy parameters into the episode-wise robot action trajectory?*
>
> * Instead of a high-dimensional representation of a trajectory (e.g. each state in a single time-step), Motion Primitives (MPs) introduce a lower dimensional vector $\theta$ which concisely defines the trajectory[2].  The generation of the trajectory depends on the method that is used. ProMPs [3]  for example define the whole trajectory as a linear function of $\theta$ as $\tau = \Phi^T \theta$, where $\Phi$ are pre-defined and time-dependent basis functions (e.g. normalized radial basis functions). Dynamic Movement Primitives (DMPs) [4]  rely on second-order dynamic systems that provide smooth trajectories in the position and velocity space. In our work, we use ProDMPs [2] which combine the advantages of both methods. In the context of reinforcement learning, the policy $\pi(\theta|c)$, or in our case, the chosen expert $\pi(\theta|c,o)$ defines a normal distribution over the parameters $\theta$ depending on the context $c$.  The agent can therefore quickly adapt to contexts and follow the trajectory for example using a PD-controller. We have added a dedicated section in Appendix B to further clarify the relationship between our work and motion primitives. If there are any additional unclarities that should be addressed, we would appreciate the feedback from the reviewer
>
>   [2] Ge Li et al, ProDMP: A Unified Perspective on Dynamic and Probabilistic Movement Primitives
>
>   [3] A. Paraschos et al, Probabilistic movement primitives
>
>   [4] Stefan Schaal et al, Dynamic movement primitives a framework for motor control in humans and humanoid robotics
>
> 2. *"Eq (3) seems to be not original from this work, a proper reference would help readers to understand the background of this line of work. The derivation from Eq (4) to Eq (5) and Eq (6) is unclear. It would be more clear to have an intermediate objective which is jointly optimizing for $\pi(\theta|c,o)$ and $\pi(c|o)$, and derive from there to have two separate objectives for bi-level optimization."*
> * We thank the reviewer for this comment. We have added the reference in the revised manuscript.
> * The derivations of the lower bounds for the expert and per-expert context distributions have been proposed and derived in detail in [5]. However, we agree with the reviewer that the derivation from Eq. (4) to Eq.(5) (now Eq.(6) in the updated paper) is not trivial. Therefore, we have included an intermediate step (now Eq.(5)) to clarify it.
>
>     [5] Celik et al, Specializing Versatile Skill Libraries using Local Mixture of Experts
>
> 3. *"In Section 3.1, it says “mapping the context to a mean vector and a covariance matrix” and “Note that in most cases a context dependent covariance matrix is not useful and introduces unnecessary complex relations.” It is confusing that whether the covariance matrix in the implementation is context dependent."*
>
> * We agree with the reviewer that this sentence might lead to confusion. Indeed, in our experiments, we have observed that using a full covariance matrix that does not depend on the context leads to more stable training behavior. We therefore did not use a context-dependent covariance matrix in our experiments. To clarify this, we have described the parameterization in Appendix A in detail.
>
> 4. *"Line 10 in Section 3.2, should “Fig. 2c” be “Fig. 2d”?"*
>
> * We thank the reviewer for pointing out the mistyped reference. We have corrected this mistake.

---

> ### Author Response · Authors · 2023-11-20
> **Reply to Reviewer w2WU continued**
>
> 5. *"Which terms in Eq (8) and Eq (9) accounts for encouraging the coverage of the context space by experts? From the formulation, it seems to try to learn a set of policy each of which can solve the entire task space as much as possible. The learning of policies seem to be relatively independent and is it possible to learn a set of experts whose preferred context distributions are the same."*
>
> * We thank the reviewer for pointing out this important question which is crucial to understanding our proposed method. Please note that the equations the reviewer was referring to are numbered Eq.(9) and Eq.(10) respectively in the updated paper.
>
>     * We first want to clarify the roles of the individual terms in Eq.(9). The objective incentivizes each expert $\pi(\theta|c,o)$ to maximize its entropy. This means that the expert should cover as much as possible of the parameter space, i.e. $\theta$- space, while still maximizing the reward $R(\theta, c)$. At the same time, the objective incentivizes the expert to cover only context-parameter regions that are not covered by other experts $o$. This is guaranteed by the variational distribution $\tilde{\pi}(o|c, \theta)$ which translates to the probability of assigning a context-parameter pair to expert $o$. As a result, the agent will try to maximize this probability as it is an argument of the log.
>
>     * A similar relation can be seen in Eq. (10). The per-expert context distribution $\pi(c|o)$ will try to cover as much as possible of the context region due to the entropy bonus. Yet, due to the variational distribution $\tilde{\pi}(o|c)$ the objective incentivizes the agent to distinguish between the different per-expert context distributions for a given context. In other words, the objective forces each $\pi(c|o)$ to concentrate more on regions that are not covered by other experts $o$ because the log of the variational distribution rewards context samples that can be assigned to $o$ with high probability.
>
>     * We updated the manuscript and clarified these relations in Section 3.3
>
>     * As a result, the optimization of the different experts and per-experts are coupled by the variational distributions which hinder them from concentrating on the same context-parameter regions. Please note that overlapping regions exist that explicitly are desired to find diverse solutions to the same task defined by context $c$. In other words, it is favorable that different experts cover a part of the same context as their adjusted Motion Primitive Parameter $\theta$ will lead to different solutions to the same context, which is incentivized by $\tilde{\pi}(o|c,\theta)$. The amount of “overlapping” in the context space depends on the choice of the hyperparameters $\alpha$ and $\beta$, which was analyzed in detail in prior work [5] already
>
> 6. *"More testing environment description would be helpful. Some details about action space and reward definitions are missing for both tasks."*
>
> * We thank the reviewer for pointing out the missing descriptions. We have added a thorough description of the environmental details in Appendix C.
>
> 7. *"Evaluating the algorithm on more environments will make the comparison more thorough. For example, it would be helpful to evaluate on the other tasks used in Otto et al. 2023."*
>
> * We agree that evaluating the algorithm on more environments is helpful. We are currently working on more evaluations and plan to provide the results before the deadline of the reviewer-author discussion phase.

---

> ### Author Response · Authors · 2023-11-20
> **Reply to Reviewer w2WU continued**
>
> 8. *"It would also helpful to show complete comparison against both SVSL and BBRL on all evaluated tasks (at least provide comparison plots in Appendix)"*
>
> * We believe that the reviewer meant the lack of comparison to SVSL only, as the results of BBRL are reported on all environments already. But please let us know if that is not the case, and please further clarify your suggestion in this case. We agree that a complete comparison to SVSL is valuable. However, due to the increased complexity of the context space, we were not able to successfully run SVSL on the extended Table Tennis and Box pushing environment. We believe there are several reasons for this: SVSL requires prior knowledge about the context space. For the table tennis environment, for example, we need to know the ball’s flying trajectory to determine whether the serving position on the robot’s table side is valid or not, in order to provide a guiding punishment term such that the per-expert context distributions are usefully updated. Alternatively, a constant punishment term could be added to the reward if the context is invalid after running in the simulator. We tried the latter and could not successfully train SVSL. Furthermore, SVSL has difficulties training experts for higher dimensional context spaces that additionally introduce non-linearities in the behavior space, for example in the box pushing task with obstacles. The Gaussian parameterization of the per-expert distributions and the linear experts were not sufficient to learn skills in our experiments.
>
>     Yet, we agree that additional comparison is useful to strengthen the method. Therefore, we have run a similar variant of SVSL, where we have replaced SVSL’s per-expert context distribution with the energy-based context distribution of Di-SkilL. Additionally, we have parameterized the experts as linear experts. The results can be seen in Fig. 4(a) and Fig.4 (b) in the updated paper.  ‘LinDi-SkilL’  overcomes the problem of the need for pre-knowledge of the context space as required by SVSL, but is not able to reach the performance of the non-linear expert counterpart Di-SkilL.
>
> 9. *"Given the proposed approach is built upon SVSL with two improvements, it would be great to do ablation study on both improvement techniques."*
>
> * We agree that such a comparison is useful to strengthen our method. The benefits of Di-SkilL over SVSL are:
>     1. Di-SkilL does not make any assumptions about the environment’s context space. I.e. due to Di-SkilL’s energy-based model (EBM) for the per-expert context distribution, there is no need to shape the reward function with guiding punishments to ensure that contexts are sampled in a valid region. This is particularly beneficial if the context influences the dynamics of relevant objects in the environment (e.g. initial ball velocity in table tennis), requiring pre-knowledge for defining a useful punishment function.
>     2. The EBM can represent hard non-linearities (visualized in e.g. Fig. 2d)) in the context space. These types of context regions are commonly present in applications, for example in the table tennis task, where the ball’s initial velocity has an upper bound and might lead to additional  “holes” in the probability space depending on the ball’s initial and target ball landing positions. These probability landscapes could also be represented with e.g. Gaussian Mixture Models as in SVSL, but require many components to achieve a reasonable approximation
>     3. Di-SkilL can learn highly non-linear experts represented by neural networks. This is beneficial for learning complex behaviors in sophisticated environments and provides a big advantage over linear experts e.g. in SVSL.
>
> * As mentioned in our answer to Question 8, we were not able to successfully train an SVSL agent on the extended Table Tennis and Box Pushing task. However, we have conducted experiments with ‘LinDi-SkilL’ which aims to prove the advantages of Di-SkilL over SVSL. ‘LinDi-SkilL’ uses the EBM of Di-SkilL, but has linear expert parameterizations. We can now clearly see that ‘LinDi-SkilL’ is able to learn useful skills (Fig. 4a) +b)), but can not reach the performance of Di-SkilL, indicating that the EBM helps shape the expert's curriculum, but the linear experts are not sufficient to perform well enough. We believe these additional experiments address the reviewer’s concerns and are open to further suggestions

---

> ### Author Response · Authors · 2023-11-20
> **Reply to Reviewer w2WU continued**
>
> 10. *"The multi-modality in this work is achieved by mixture of experts, however each expert is still modeled by uni-model gaussian policy."*
> * It is true that each expert is modeled as a uni-model Gaussian, however, as described in the answer to Question 5, the overlapping probability mass to a certain extent in the context space enables learning diverse, i.e. multi-modal behaviors. For these overlapping context regions several experts $\pi(\theta|c,o)$ are responsible and provide different MP parameters $\theta$ which lead to different behaviors. This is also visualized in in Fig. 4 c)- e), Fig. 5 as well as in the videos in the supplementary material.
>
> 11. *"Recent work (Huang et al. 2023 [1]) proposes some multi-modal policy parameterization. How is the proposed approach compare to this work and can the proposed approach enhanced by the policy reparameterization from [1]?"*
> * We thank the author for mentioning the work in [1]. The proposed objective in [1] is related to the objective we use. We have therefore included this work in the related work discussion.  Yet, we want to emphasize that the objective shown in Section 2 was proposed in prior work [5]. Both objectives are similar in the sense that they lower bound an initial objective using techniques from variational inference for usage with latent variable models. Additionally, if the objective in [5] is applied to the classical MDP, i.e. step-based RL set up without any curriculum learning (learning the gating $\pi(o|c) $ instead of $\pi(c|o)$), the resulting lower-bound is the same as proposed in [1].
>
>     However, there are significant differences to our setting: First, the work in [1] does not consider curriculum learning, i.e. there is no context distribution learned, the work does not consider the Contextual Policy Search problem using Motion Primitives (MPs), i.e. their work act under the commonly known Markov decision process assumptions. Furthermore, we consider a Mixture of Experts policy with a discrete number of experts which does not require parameterizing the responsibility ($\pi(o|c,\theta)$) and the gating ($\pi(o|c))$ terms as they can be calculated in closed form. Finally, the work in [1] considers the Model-based Reinforcement Learning case where we do not learn a dynamics model.
>
>     Indeed, considering a continuous latent variable o (z in [1]) is an interesting approach for future work and could improve the performance
>
> 12. *"Is this proposed approach also applicable to step-based RL problems?"*
>
> * Applying our approach to step-based RL is an interesting research question that requires detailed analysis especially how to integrate the learning of the context distribution that is responsible for automatic curriculum learning. Probably the most intuitive option would be to allow the agent to set the context once at the beginning of the episode and then consider the learning problem in the standard MDP framework. Both frameworks have their advantages and disadvantages. While step-based approaches exploit the temporal structure of RL problems and are therefore expected to outperform episode-based RL in terms of sample efficiency, episode-based RL methods are usually sample inefficient but are able to smoothly explore in trajectory space which leads to improved performance in sparse and non-markovian reward settings over step-based approaches [6]. In this work, we focused on developing a new method in the framework of episode-based RL and consider the application of the approach in the step-based RL framework as an interesting future work.
>
>     [6] Otto et al, Deep Black-Box Reinforcement Learning with Movement Primitives

---

> ### Comment · Reviewer_w2WU · 2023-11-21
> **Re: Authors' Response**
>
> Thanks for providing a detailed response to the review. The authors' explanations and the updated manuscript address most of my questions. My main remaining reservation is the fact that the current proposed approach is limited to episodic policy scenarios which could constrain the impact of the proposed approach on more practical control problems. I also appreciate that the authors promise to run additional experiments on more benchmark problems. I'm willing to increase my score if the authors can show more convincing results on the other benchmark problems from [Otto et al. 2023]

---

> ### Author Response · Authors · 2023-11-22
> **RE: Reviewer's Response**
>
> We thank the reviewer for their response. We want to clarify that contextual episodic policy search (CEPS) is also applicable to control problems where step-based algorithms are applicable too. In contrast, there are scenarios, such as environments with non-markovian rewards, to which common step-based RL methods are not applicable due to the Markov Decision Process (MDP) assumption. CEPS will usually explore in the parameter space of the trajectory generator and track this generated trajectory with a PD-controller. The disadvantage of these methods is that they usually can not react to e.g. unplanned collisions with other objects. However, this is an interesting approach that we aim to investigate in future work (see Section 5).
>
> We would like to inform the reviewer that we have
> * added a comparison to PPO in Fig. 4 b)
> * added a 5-Link Reacher benchmark with sparse rewards in time (Fig. 3a) + Fig. 3c)
> * added a challenging robot mini golf benchmark with non-markovian rewards (Fig. 3a) + Fig. 4c)
>
> We would like to note that the reacher benchmark is different as opposed to the benchmark presented in Otto et al 2023 in that we consider the whole context space as possible goal-reaching locations. In their proposed environment, the authors consider only the upper half plane (first and second ) to avoid multi-modalities in the solution space. Therefore, considering the whole context space makes the task harder in general and explains why BBRL performs differently.
>
> All details regarding the environments are listed in Appendix C. However, if the reviewer has further open questions, we are happy to answer them.

---

### Official Review · Reviewer_vLLv · 2023-10-31

**Soundness:** 2 fair
**Presentation:** 2 fair
**Contribution:** 2 fair
**Rating:** 6
**Confidence:** 3

**Summary:**

The paper introduces Di-SkilL, a reinforcement learning (RL) approach for training agents to exhibit multi-modal and diverse skills. The authors propose a mixture of experts (MoE) model that enables the agent to select and adapt from a repertoire of skills based on the context. The context in this work represents task definitions like goal positions or varying environmental parameters. The authors leverage energy-based models for per-expert context distributions to overcome challenges in multi-modality representation and hard discontinuities. They demonstrate the efficacy of their approach on complex robot simulation tasks.

**Strengths:**

The paper addresses an important and timely challenge in RL, that of equipping agents with the ability to learn and adapt to multiple skills for a given task. The energy-based approach for representing per-expert context distributions is innovative and offers a solution to traditional Gaussian parameterization limitations. The model's design, which avoids assumptions about the environment and doesn't require prior knowledge, increases its general applicability.

**Weaknesses:**

There might be concerns regarding the scalability and computational efficiency of the proposed method, especially in real-world robotic applications. This should be discussed.

Related work discussion and baseline are not sufficient, missing other MoE methods like PMOE [1].

[1] Ren, Jie, et al. "Probabilistic mixture-of-experts for efficient deep reinforcement learning." arXiv preprint arXiv:2104.09122 (2021).

**Questions:**

See Weakness.

---

> ### Author Response · Authors · 2023-11-22
> **Reply to Reviewer vLLv**
>
> We would like to thank the reviewer for the insightful comments and the dedicated time invested to review our work. We would like to clarify the reviewer’s concerns and questions in the following. We have updated the paper and marked the changed text motivated by the suggestions of the reviewer in violet.
>
> *"There might be concerns regarding the scalability and computational efficiency of the proposed method, especially in real-world robotic applications. This should be discussed."*
> * Computational Efficiency:
>     * Our method considers a more powerful parameterization of the policy than commonly used Gaussian policies. Therefore, it is natural, that this requires more intensive computation compared to single, uni-modal policies. However, we want to clarify in the following, why the proposed framework still allows for a computationally efficient training process.
>         * CEPS considers ($\theta$, $c$, $R$) tuples for updating the policy. As a single $\theta$ corresponds to a single trajectory, CEPS generally allows for a very efficient sampling process by high parallelization possibilities. In fact, this is easily realizable with already existing parallelized sampling environments for example as provided by  Gymnasium [2].
>         * We can apply this concept easily to our approach, by first sampling the contexts $c \sim \pi(c|o)$ and then sampling the parameters $\theta \sim \pi(\theta|c,o)$ for each $o$  and finally execute these samples in parallel. This allows for efficient usage of the available resources.
>         * Please note that on-policy algorithms in which CEPS is included can parallelize sampling in general, whereas this is not efficiently realizable for off-policy methods such as SAC [3] as they do not generate enough samples within one iteration.
>         * Furthermore, the used objective allows for updating each component and its corresponding context distribution independently. Therefore, updating these models can be efficiently parallelized. As we use rather small networks for the experts we can easily perform the updates on the CPUs.
>         [2]: Gymnasium
>
>         [3]: Haarnoja et al, Soft actor-critic: Off-policy maximum entropy deep reinforcement learning with a stochastic actor
> * Scalability:
>     * We would address the remark with the following explanation, but would also appreciate it if the reviewer could confirm if this answers their question, or clarify the remark. In general, we parameterize our experts and per-expert distributions with neural networks which are known to scale favorably to higher dimensions. Please note that we have 25+ dimensional parameter spaces in our experiments, which is already a rather high-dimensional space for robot control tasks — Details regarding the parameter, and context spaces are described in the updated manuscript in Appendix C.
> * Real-world robotic applications:
>     * We agree that the direct application of Di-SkilL on real-world applications is hard to determine. However, RL with Motion Primitives (MPs) has a high potential for successful sim2real transfer.
>         * Reinforcement Learning algorithms are generally difficult to deploy in real-world experiments due to the high sample complexity, especially in robot manipulation tasks. Additionally, many RL algorithms, especially step-based RL methods, result in very jerky random walk behavior due to their exploration strategy (random action noise). This behavior can damage the robot [4]. Finally, obeying the safety constraints of the real-world system within the reinforcement learning algorithm is an open and ongoing research question [6]. This is an additional interesting direction of future work worth examining
>         * While Motion Primitive RL (including Di-SkilL) methods lead to smooth, efficient, and time-correlated exploration behavior and therefore can handle jerky random walks [4], they still suffer high sample complexity and the problem of considering safety constraints. We believe that the sample complexity can be efficiently addressed by off-policy versions of our method. However, we consider tackling this problem as an interesting future work (as mentioned in Section 5).
>         * Yet, movement primitives have been shown to be directly applicable for sim2real transfer, e.g. [5], if good tracking controllers on the real robot are available. Consequently, for real-world applications, we would first train in simulation and then adapt the policy to the real-world setting if necessary.
>
>         [4] Otto et al, Deep Black-Box Reinforcement Learning with Movement Primitives
>
>         [5] Self-Paced Contextual Reinforcement Learning
>
>         [6] Javier Garcia, A Comprehensive Survey on Safe Reinforcement Learning.
>
>     If our answers address the reviewer's concerns we are happy to include them in the paper.

---

> ### Author Response · Authors · 2023-11-22
> **Reply to Reviewer vLLv continued**
>
> *"Related work discussion and baseline are not sufficient, missing other MoE methods like PMOE [1]."*
>
> * We thank the reviewer for pointing out the work by [1]. We have extended our discussion on related works and included PMOE. Regarding the comparison to PMOE as a baseline, we would like to note that PMOE requires the Markov assumptions, as it relies on step-based RL methods like SAC, or PPO. Given that Di-SkilL is categorized in the contextual episodic reinforcement learning framework and that we train in environments with non-markovian rewards, we do not expect PMOE to perform favorably in these environments. However, we could add a comparison to Di-SkilL on the extended Box Pushing task, but won’t be able to do that within the author-reviewer discussion face due to missing computational resources. We will add the comparison to the camera-ready version though. Please note that we have added ‘LinDi-SkilL’ as presented in the answers to Reviewer w2WU as an additional baseline. Additionally, we plan to add PPO on the extended Box pushing task as well as another evaluation of an environment.

---

> > ### Comment · Reviewer_vLLv · 2023-11-22
> >
> > Good, I would like to raise my score.

---

> > > ### Author Response · Authors · 2023-11-22
> > > **RE: Comment by Reviewer vLLv**
> > >
> > > We thank the reviewer for their response and for increasing the score of the paper.

---

### Official Review · Reviewer_BduN · 2023-11-05

**Soundness:** 2 fair
**Presentation:** 3 good
**Contribution:** 3 good
**Rating:** 6
**Confidence:** 3

**Summary:**

This paper proposes an approach for acquisition of diverse skills using non-linear mixture of experts. The main ingredients of this approach are maximum-entropy objective for learning diverse experts, trust-region optimisation for stable bi-level optimisation, and energy-based models for automatic curriculum learning. Their approach demonstrates the learning of diverse skills for solving the same task.

**Strengths:**

- Section 3 on Diverse Skill Learning is well-written and describes the method and the contributions of the work in a clear manner, with the appropriate references to existing work in the area.
- Figure 5 provides good qualitative evidence of diverse skills being learnt by the proposed approach.
- The Conclusions mention a drawback of the approach in that it is unable to replan in the event of collisions, for instance. This is an important empirical detail and I liked the fact that it was raised in the paper.

**Weaknesses:**

- Automatic curriculum learning is a key ingredient of the proposed method; however, an important set of approaches in this direction has not been covered in related work, such as [1] and others in this family of approaches.
- Figure 3-c which shows ablations on the TT environment has inconsistent number of episodic samples (X-axis) for the different approaches in the plot. It would be useful to have asymptotic performance of each of these approaches and then compare them in terms of this performance, and also in terms of training speed (eg: w/o automatic curriculum learning is slower than w/ automatic curriculum learning).
- In Figure 4 a-b as well, it would be nice to have the asymptotic performance for Di-Skill and BBRL to have a fair comparison of performance.
- While SVSL and BBRL are good CEPS baselines, it would also be nice to compare with a standard RL baseline such as PPO to better motivate the need for this approach.
- Minor points:
    - The environment description has been duplicated to some extent in the main text and the caption for Figure 4. It may help to prune that and instead include additional analysis.
    - Section 2 Prelimanaries -> Preliminaries

[1] Prioritized Level Replay. Jiang et al, 2020.

**Questions:**

- I am not sure why the prior $\pi(o)$ has been assumed to be uniform. For sparse reward tasks, I can imagine observations that are closer to the goal would be rarer than those closer to the initial state at the beginning of the episode. Or does this paper assume full access to the simulator in which resetting to any observation is possible?
- In the Experiments section, the paper mentions that the aim is to check whether Di-Skill is able to learn precise and diverse skills. The fact that it learns diverse skills is reasonably demonstrated in Figure 5, but I am yet to find evidence that precise skills are learnt. Could the authors please point me to that?
- Could the authors provide any insights on the learnt $\pi(o | c)$? Of the number of experts used, how often were they used when averaging across observations?

---

> ### Author Response · Authors · 2023-11-21
> **Reply to Reviewer BduN**
>
> We would like to thank the reviewer for the insightful comments and the dedicated time invested to review our work. We would like to clarify the reviewer’s concerns and questions in the following. We have updated the paper and marked the changed text motivated by the suggestions of the reviewer in orange.
>
> *"Automatic curriculum learning is a key ingredient of the proposed method; however, an important set of approaches in this direction has not been covered in related work, such as [1] and others in this family of approaches."*
>
> * We appreciate the reviewer’s advice. We have added a set of works in this family of approaches to the related work discussion
>
> *"Figure 3-c which shows ablations on the TT environment has inconsistent number of episodic samples (X-axis) for the different approaches in the plot. It would be useful to have asymptotic performance of each of these approaches and then compare them in terms of this performance, and also in terms of training speed (eg: w/o automatic curriculum learning is slower than w/ automatic curriculum learning)."*
>
> * We thank the reviewer for this comment. The goal of this ablation is to show that automatic curriculum learning is a necessary feature of Di-SkilL to learn useful and high-performing skills. Hence the main focus of Fig. 3 c) is the comparison of Di-Skill, Di-SkilLwoCurrV1, and Di-SkilLwoCurrV2. All of the variants are already converged at different numbers of samples. While Di-Skill and Di-SkillwoCurrV1 converge at a similar number of samples, they differ in performance (around 20% difference). Di-SkilLwoCurrV2 needs around twice the number of samples to reach a similar success rate as Di-SkilL.
> Yet, we agree that the learning curve of SVSL was not fully converged. We have rerun SVSL with improved hyperparameters and updated the results in Fig. 3 c). Although there is an improvement in the convergence speed, the performance at the end of the training is the same as before.
>
> *"In Figure 4 a-b as well, it would be nice to have the asymptotic performance for Di-Skill and BBRL to have a fair comparison of performance."*
>
> * We have rerun both algorithms for an increased number of samples and updated the results in Fig. 4, such that the asymptotic performance of both methods can be clearly compared. Di-SkilL outperforms BBRL by a small margin on the extended table tennis task and clearly achieves better performance (around 20% better) on the Box Pushing task. Please note that we have rerun Di-SkiLL and BBRL with full covariance parameterization on the Box Pushing task, which helped increase the performance for both methods.
>
> *"While SVSL and BBRL are good CEPS baselines, it would also be nice to compare with a standard RL baseline such as PPO to better motivate the need for this approach."*
>
> * We agree with the reviewer that adding PPO as a baseline emphasizes the benefits of Di-SkilL. We are currently working on running PPO on the Box Pushing task, as this is the only environment satisfying the Markov assumption of PPO. We will be adding the results before the end of the reviewer-author discussion deadline.
>
>     We were not able to successfully run SVSL on the extended table tennis and box pushing tasks, as SVSL requires to design of a punishment term for guiding the per-expert context distribution (e.g. distance to the valid context regions). Designing a suitable punishment term requires good reward-shaping experience and in most cases, knowledge about the influence on the physics of the environment. For example, even though the ball’s initial velocity in the table tennis task is bounded by box constraints, only specific constellations with the other context dimensions are physically valid, leading to a complex probability landscape of the valid region. If the ball is initialized with a high velocity, it might happen that the desired ball landing position is not met. Since the per-expert context distributions define the contexts at the beginning of the episode, it can easily happen that non-valid contexts are sampled in the case of SVSL. However, to account for the missing comparison to SVSL, we have added “LinDi-SkilL”, a variant of Di-SkilL with linear experts. “LinDi-SkilL” benefits from the energy-based model from Di-SkilL and hence does not need special treatments for learning the per-expert context distribution. Additionally, we used linear experts as in SVSL. We have run all experiments for 24 seeds and report the results in Fig.4. For the same number of training samples, LinDi-SkilL can not achieve the performance of Di-SkilL, indicating the advantages over SVSL.

---

> ### Author Response · Authors · 2023-11-21
> **Reply to Reviewer BduN continued**
>
> *"The environment description has been duplicated to some extent in the main text and the caption for Figure 4. It may help to prune that and instead include additional analysis."*
>
> * We thank the reviewer for this suggestion to improve our paper. We have updated the text to the environment description in the main text. A detailed description of the environments can be found in Appendix C.
>
> *"I am not sure why the prior $\pi(o)$  has been assumed to be uniform. For sparse reward tasks, I can imagine observations that are closer to the goal would be rarer than those closer to the initial state at the beginning of the episode. Or does this paper assume full access to the simulator in which resetting to any observation is possible?"*
>
> * We believe there is a misunderstanding: The distribution $\pi(o)$ is a discrete distribution, specifying the probability of choosing an expert $o$ without observing a context $c$, Thus, $o$ is not the observation of the environment. Generally, the prior can be interpreted as the preference of specific experts $o_{1}, o_2, ...$. This means that the user could include an inductive bias by setting a higher probability to preferred experts. However, due to the gating distribution $\pi(o|c)=\frac{\pi(c|o)\pi(o)}{\pi(c)}$ the agent still decides which expert $o$ is suitable based on the current context $c$. Please note that the prior $\pi(o) $ can be freely chosen. We set it to be uniform in our work, because it is unclear before the training which expert should be preferred and because a uniform distribution over the experts maximizes the entropy (see Eq. (5) in the updated version of the paper.) However, we agree that the sentence “The prior $\pi(o)$ is assumed to be a uniform distribution throughout this work” might lead to confusion, that why we changed it to “The prior $\pi(o) $ is set to a uniform distribution throughout this work.
>
>     Based on the follow-up question of the reviewer we believe they meant the context $c$, as the context can be interpreted as the observation in the step-based RL setting.  In our work, we implement automatic curriculum learning by allowing the agent to choose its favored contexts $c$ and set them in the environment.  Favored contexts are sampled from the per-expert context distribution $\pi(c|o)$.  Consequently, we assume that the agent can set the context $c$ once at the beginning of the episode. This is done during training such that each expert is allowed to discover the context regions it favors. Please note that even though the optimization of each expert is independent, they are still coupled via the variational distributions $\tilde{\pi}(o|c,\theta) $ and $\tilde{\pi}(o|c)$ such that preferred context regions that are covered by other experts won’t be attractive to the current agent anymore (Eq. (6) + Eq. (7)). During inference, the agent observes a context $c$ and chooses the expert which then executes the adjusted motion primitive parameter $\theta$.
> * We would appreciate it if the reviewer could confirm whether our explanations answer their question, or clarify the remark.
>
> *"In the Experiments section, the paper mentions that the aim is to check whether Di-Skill is able to learn precise and diverse skills. The fact that it learns diverse skills is reasonably demonstrated in Figure 5, but I am yet to find evidence that precise skills are learnt. Could the authors please point me to that?"*
>
> * By precise skills, we meant, whether the agent is able to learn high-performing skills. The provided environments require highly precise motions to successfully solve the task. In table tennis, for example, the task is considered successful if the distance between the ball’s landing position and the goal position is below 0.2m. This requires the robot to precisely hit the ball, as displaced hitting will lead to different landing positions. In the Box Pushing task, the agent needs to drag the box such that the distance error of the box’ end and goal position is below 0.05m and the rotation error is below 0.5rad, while dealing with e.g. unknown dynamics and friction. However, to avoid confusion we have replaced the term ‘precise’ with ‘high-performing’ which we believe is more intuitive. If the reviewer has other suggestions, we are happy to include them in the description.

---

> ### Author Response · Authors · 2023-11-21
> **Reply to Reviewer BduN continued**
>
> *"Could the authors provide any insights on the learnt $\pi(o|c)$ ? Of the number of experts used, how often were they used when averaging across observations?"*
>
> * the gating distribution is not explicitly parameterized and is indirectly learned via
>
>     $\pi(o|c)=\frac{\pi(c|o)\pi(o)}{\sum_{o'}\pi(c|o')\pi(o')}~~~~~~~~(1)$
>
>     as $\pi(c|o)$ is a learned energy-based model. During inference, the agent observes a context $c$, which is used to calculate the gating with equation (1) resulting in probabilities of the different experts. Based on this probability, an expert $o$ is sampled and used to sample a motion primitive parameter $\theta \sim \pi(\theta|c,o)$. During inference, i.e. testing, we sample only one expert per context. If the user is interested in generating diverse skills (such as in Fig. 4 and Fig. 5, and the video in the supplementary material), the user can sample several times from $\pi(o|c)$, depending on how many different skills should be executed.

---

> > ### Comment · Reviewer_BduN · 2023-11-22
> >
> > Thanks for the detailed responses to my questions, running additional experiments, and also for the clarifications. I have increased my score.

---

> ### Author Response · Authors · 2023-11-22
> **Re: Comment by Reviewer BduN**
>
> We thank the Reviewer for their response and for increasing the score of our paper. We have now added the comparison to PPO (Fig 4b) and 2 additional benchmarks (5-Link Reacher and Robot Mini Golf -- Fig 3a) + Fig 3c) + Fig 4c)).

---

### Author Response · Authors · 2023-11-20
**Revised Version**

Dear Area Chair and reviewers,

currently, we are working on incorporating the suggestions of the reviewers into our manuscript and would like to provide a changelog as we will have different versions during the reviewer-author discussion period. For easy reference, we color-code all changes in the manuscript depending on the reviewer who suggested them:
* Reviewer  BduN: orange

* Reviewer vLLv: violet

* Reviewer w2WU: green

* Reviewer ydwi: blue

Additionally, we provide a brief changelog and in the parenthesis, we denote the reviewer whose comments motivated these changes:

* [17.11.2023]:
    * Improved motivation in  Section 1 (Introduction), Section 2 (Mixture of Experts (MoE) Policy for Curriculum Learning (ydwi)
    * Detailed description of the parameterization of the Mixture of Experts (MoE) model in Appendix A (ydwi)
* [20.11.2023]:
    * Extended discussion on related works in the field of Curriculum Reinforcement Learning in Section 2 (BduN)
    * Extended description of the objective in Section 2 (w2WU)
    * Extended description of related works in the field of Diverse Skill Learning in Section 2 (vLLv and w2WU)
    * Extended description of the individual terms responsible for learning diverse solutions and coverage of the context space in Section 3.3  (w2WU)
    * Additional details on the usage of Motion Primitives in Reinforcement Learning in Appendix B (ydwi, w2WU)
    * a thorough description of the tasks in Appendix C (w2WU, ydwi, BduN)
    * New results for SVSL in Section 4, Fig. 3 c): Improved hyperparameters and longer runs (BduN)
    * New results for Di-SkilL, BBRL on the extended table tennis (Fig. 4a))  and the extended box pushing tasks (Fig. 4b)). We have rerun both algorithms longer for improved comparability (BduN) and found improved better-performing hyperparameters for Di-SkilL and BBRL on the box pushing task.
    * Added LinDi-SkilL as an alternative comparison to SVSL on the extended table tennis task (Fig. 4a)). (w2WU). We will provide the results for the extended box-pushing task
* [21.11.2023]:
    * Added LinDi-SkilL as an alternative comparison to SVSL on the extended box pushing task (Fig. 4b)). (w2WU).
* [22.11.2023]:
    * Added a dedicated discussion on the learned Mixture of Experts Model trained with Di-SkilL in Appendix D (ydwi)
    * Added comparison to PPO on Box Pushing task (BduN)
    * Added 5-Link Reacher task with full context space as apart to only non-multimodal, half-plane space presented in Otto et al 2022 (ydwi, w2WU)
    * Added challenging robot mini golf benchmark (ydwi, w2WU)
    * Added detail description of new environments in Appendix C (w2WU, ydwi, BduN)

---

### Meta-Review · Area_Chair_UsJw · 2023-12-12

**Metareview:**

The paper proposes a reinforcement learning (RL) framework (Di-Skill) for learning diverse, multimodal skills. Underlying the framework is a mixture of experts-based representation of the policy, where each "expert" represents a skill as a contextualized motion primitive. To encourage the policy to express a diverse set of skills, the framework combines an energy-based per-expert context distribution and a maximum entropy objective. The paper evaluates the method alongside baselines on simulated robot control tasks.

The paper was reviewed by four reviewers, each of whom read and replied to the authors' responses. The reviewers appreciate the importance of learning diverse skills in RL and find the means by which the paper proposes learning these skills to be intuitive. The reviewers initially raised concerns regarding the experimental evaluation, both in terms of the limited number of domains that were considered and the way in which the results were presented; and the placement of this work, particularly in the context of work on mixtures of experts. The authors made a considerable effort to address these concerns by re-running experiments to clarify the presentation of results, running experiments on new benchmarks, and updating the paper to more clearly describe the framework and its relation to existing work. As a result, three of the reviewers increased their overall recommendations, while the fourth reviewer decided to maintain their score due to what they see as a need for additional benchmark evaluations (e.g., Meta-World, Hopper, and Beer Pong from Otto et al. 2023, the lack of which the review notes was not addressed in the author response).

The paper has the potential to provide a valuable contribution to skill learning for RL. While the paper has improved significantly as part of the review process, it would benefit from a more thorough experimental evaluation that includes a broader suite of domains.

**Justification For Why Not Higher Score:**

More experiments are necessary. None of the reviewers are particularly excited about this work.

**Justification For Why Not Lower Score:**

N/A

---

### Decision · Program_Chairs · 2024-01-16

Reject